# SCRIBE: Stroke- and Context-Regularized Test-time Adaptation for Handwritten Text Recognition

## Abstract

Handwritten text recognition (HTR) converts images of handwritten text—from lines to full pages—into accurate, machine-readable transcriptions. However, it often operates under distribution shift—new writers, historical substrates, scanning artifacts, layouts, and even cross-language use—precisely when target labels and source data are unavailable. Although recent foundation models perform well on their training distributions, their generalization across domains is fragile. Limitations in capacity, inadequate pretraining scale, or corpus–domain mismatch frequently lead to pronounced errors, underscoring the need for efficient adaptation even with state-of-the-art pretrained models. We fill this gap by adapting a foundation model at test-time without labels or source data. To the best of our knowledge, this is the first HTR test-time adaptation approach that jointly optimizes a lightweight stroke-structure loss with a document-conditioned language prior, rather than treating linguistic (LM decoding/reranking) and visual (self-training/normalization) cues separately. Evaluated on four benchmarks (George Washington, IAM, RIMES, Bentham), our approach achieves an average absolute reduction of **0.0341** in CER and **0.0427** in WER, corresponding to mean relative improvements of **20.8%** and **12.8%**, respectively. These findings demonstrate that integrating lightweight visual and linguistic priors provides an effective strategy for test-time adaptation in HTR.

## 1 Introduction

Handwritten text recognition (HTR) converts handwriting in scans or photos into machine-readable text. Despite advances in model architectures (Puigcerver, 2017; Michael et al., 2019; Coquenet et al., 2022; d'Arce et al., 2022; Li et al., 2025b; Kang et al., 2022; Barrere et al., 2022; 2024), performance still degrades under deployment-time distribution shifts, arising from unseen writers, inks, scanning artifacts, layout changes, or cross-language content (Garrido-Munoz & Calvo-Zaragoza, 2025). Recently, foundation models have advanced HTR by coupling powerful vision encoders with sequence decoders pretrained on large and diverse corpora (Kim et al., 2022), providing strong zero-shot transferability (Bautista & Atienza, 2022). Yet, without explicit exposure to unseen target distributions, their performance under real-world shifts remains limited (Chi et al., 2024).

Supervised adaptation is a common approach to mitigate distribution shift; however, privacy and licensing constraints often prevent sharing labeled data across domains, limiting its practicality in real deployments (Li et al., 2024). Test-time adaptation (TTA) offers a weaker assumption, updating the recognizer using only the unlabeled test stream (Wang et al., 2020). Recent TTA-based HTR approaches demonstrate strong personalization and stability under writer variation (Gu et al., 2025a;b). Yet these methods are typically evaluated in *in-domain* or *within-collection* settings, where the target data remain close to the training distribution and language, and variability is largely confined to writer style (Gu et al., 2025a). In contrast, production systems must operate under more challenging *out-of-domain* (OOD) conditions, where shifts extend beyond writers to new languages, layouts, or acquisition artifacts (Garrido-Munoz & Calvo-Zaragoza, 2025). Furthermore, they require customized offline training with architectural modifications and access to source data.

In this work, we build on the foundation model TrOCR (Li et al., 2023), which has demonstrated strong OOD generalization and cross-task transferability (Bautista & Atienza, 2022). We propose a test-time adaptation framework that updates TrOCR on-the-fly using only unlabeled OOD inputs,

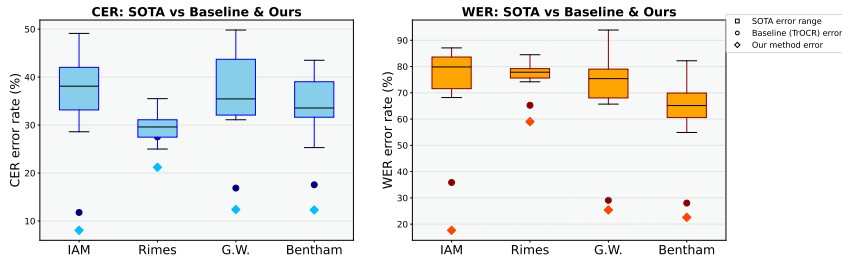

Figure 1: **Comparison of CER (left) and WER (right) on four datasets.** The foundation model TrOCR shows strong OOD generalization, and our method consistently improves upon it. **IAM** results use **TrOCR-base-stage1(pretrained-only)**; **RIMES**, **GW**, and **Bentham** use the **TrOCR-base** backbone.

mitigating distribution shift in a strictly *source-free* setting. This raises a central question: what *signals* can reliably guide adaptation under such constraints? We focus on the two deployment-time signals that remain accessible: **(i)** *visual stroke geometry* from the input image (Shit et al., 2021; Shi et al., 2024; Acebes et al., 2024; Kirchhoff et al., 2024; Hu et al., 2019), and **(ii)** *document-level linguistic regularities* from the text stream (Scheidl et al., 2018; Tarride et al., 2024; Kang et al., 2021; Tomeh et al., 2013; Wick et al., 2022; Tula et al., 2023). Prior work typically explores these two directions in isolation: visual regularizers are applied without language constraints, while LM fusion or reranking often assumes a fixed optical encoder (Tula et al., 2023). Under strict TTA constraints (unlabeled target data, no source replay and offline training), such single-channel updates leave complementary errors uncorrected (Xiao & Snoek, 2024; Zhao et al., 2023).

This motivates a coupled design that grounds the encoder in stroke geometry and the decoder in document-conditioned linguistic regularities. We instantiate this design on the foundation recognizer TrOCR (Li et al., 2023). The key idea is to leverage both sides of the encoder–decoder architecture, visual stroke geometry and linguistic token regularities, to achieve stable adaptation without labels or source replay. Concretely, we propose three principles. **(i) Geometry preservation.** The vision encoder is regularized with a Dice-style skeleton loss, augmented by a Chamfer-based shape term computed directly from each input line. This promotes centerline continuity and stroke alignment that are invariant to writer and scan variability. It also curbs topology-breaking drift common in label-free updates. **(ii) Semantic anchoring.** The decoder is stabilized via a document-conditioned KL regularizer to an external LM prior. This softly aligns token distributions under the same context. The KL term is uncertainty-gated, which reduces brittleness compared to hard pseudo-labels in OOD settings. **(iii) Decoding-time regularization.** LM rescoring of $n$-best beam hypotheses (sequence-level shallow fusion) suppresses skip/repeat errors and resolves local optical confusions that the encoder alone cannot correct.

We validate our approach on four benchmarks that span distinct axes of distributional variation. As shown in Fig. 1, our method consistently improves over the strong TrOCR across all benchmarks, yielding average relative error reduction of 27.65%, 27.65%, 23.08%, and 23.08%. These improvements appear in both character- and word-level accuracy, without inducing a trade-off between them. To the best of our knowledge, this is the first work to explicitly study *source-free* HTR—test-time adaptation on unlabeled, OOD inputs without access to source data or stored statistics. Even without labels or source replay, our framework reverses much of the accuracy drop under distribution shift, yielding strong cross-domain performance (Puigcerver, 2017; Coquenet et al., 2022; d'Arce et al., 2022; Li et al., 2025b; Kang et al., 2022; Michael et al., 2019; Barrere et al., 2022; 2024), all without retraining. Our contributions are as follows:

- **Source-free TTA for HTR.** We present the first study of *strict* test-time adaptation for HTR, updating models online on unlabeled target data without source replay or target-only fine-tuning, explicitly addressing OOD shifts in script, noise, layout, and language.

- **Minimal and deployable.** Our ~~plug-and-play~~ framework builds on TrOCR (Li et al., 2023), adds only lightweight heads with few inner steps, and adapts per document or in small batches.

- **Coupled geometry and semantics.** We regularize the encoder with Dice- and Chamfer-based stroke geometry, and stabilize the decoder with a document-conditioned LM-guided KL loss and $n$-best LM rescoring.

- **Robust OOD gains.** Our method yields consistent CER and WER reductions across benchmarks.

## 2 RELATED WORK

**Test-time adaptation for HTR.** Recent work has begun to tailor test-time adaptation (TTA) to handwriting. For text-line recognition, (Tula et al., 2023) introduces an unsupervised, source-free TTA that iteratively self-trains the optical model with language-model feedback, improving robustness across scripts and corruptions, but does not operate in exactly the same line test-time adaptation setting that we consider. Earlier HTR efforts explored unsupervised target adaptation for writer personalization without labels (Kang et al., 2020), while meta-learning–based writer adaptation (MetaHTR) performs few-shot, label-based updates at inference (Bhunia et al., 2021). In visual document understanding (VDU) settings, (Ebrahimi et al., 2022) proposes DocTTA to adapt VDU models via masked visual–language modeling and pseudo-labeling. More recently, for handwritten documents, (Gu et al., 2025b) performs test-time training using a meta-auxiliary self-supervised loss, and (Gu et al., 2025a) shows parameter-efficient personalization from a few *unlabeled* test-time examples via prompt tuning, but neither of these papers is source-free. Related TTA ideas also appear on the post-OCR side, where (Guan et al., 2024) adapts seq2seq correctors at test time.

**Skeleton-aware optimization.** Skeleton/topology losses—first used for tubular anatomy in medical imaging—have been widely applied to thin, curvilinear structures in remote sensing and document/road/crack extraction (Shit et al., 2021; Hu et al., 2019). clDice enforces soft-skeleton overlap (with cbDice and clCE variants), while persistent-homology and Skeleton Recall losses promote global connectivity with lower cost (Shi et al., 2024; Acebes et al., 2024; Hu et al., 2019; Kirchhoff et al., 2024). In document vision, centerline cues already guide robust detection (TextSnake, CRAFT) by reducing sensitivity to stroke width and clutter (Long et al., 2018; Baek et al., 2019). For HTR, characters are narrow strokes whose identity (loops, junctions, crossings) is inherently topological (Kong & Rosenfeld, 1989); importantly for source-free TTA, skeleton targets $S^\star$ are obtainable on-the-fly via fast thinning (Zhang & Suen, 1984). We therefore couple a Dice-style connectivity term with a Chamfer/shape term to align predicted centerlines $\hat{S}$ to $S^\star$, following distance-aware regularization (Karimi & Salcudean, 2019).

**Language-model guided KL and reranking for HTR.** LM guidance in HTR has been explored both at decoding—and, less commonly, as a distributional regularizer for test-time adaptation (Tarride et al., 2024; Tula et al., 2023). For CTC systems, constrained search with word/character $n$-gram LMs—e.g., Word Beam Search—yields consistent gains by injecting lexical and syntactic priors during decoding (Scheidl et al., 2018). Recent open-source ATR toolkits (e.g., PyLaia) show that even compact $n$-gram LMs, used with auto-tuned shallow fusion, reliably improve recognition and come with practical, auto-tuned recipes for deployment (Tarride et al., 2024). Beyond $n$-grams, (Kang et al., 2021) introduces *Candidate Fusion*, which integrates a neural LM into a seq2seq HWR model and jointly learns to correct typical optical errors. Reranking has also been explored: (Tomeh et al., 2013) uses linguistic/semantic features to rerank $n$-best OCR/HTR hypotheses (including handwritten Arabic), and (Wick et al., 2022) rescores seq2seq decodings with auxiliary scores to reduce skipped/repeated words. A recent *Pattern Recognition* study reports that rescoring HTR $n$-best lists with pretrained neural LMs yields additional CER/WER gains beyond $n$-gram fusion and helps curb repeat/skip errors (Li et al., 2025a).

## 3 PRELIMINARIES

**Problem setting.** We study test-time adaptation (TTA) for handwritten text recognition (HTR) (Sun et al., 2020; Wang et al., 2020) under deployment shifts such as new writers, substrates, scanners, layouts, or languages. Let $X \in \mathbb{R}^{H \times W}$ denote a line image and $s = (s_1, \ldots, s_T)$ a token sequence over vocabulary $V$. A pre-trained recognizer with parameters $\theta$ defines $p_\theta(s \mid X)$. At deployment, *neither source pre-training data nor target labels are available.* Incoming pages are segmented into line crops $\{X_i\}_{i=1}^m$, which we process in small document-level batches $\mathbb{B}$. The objective is to adapt the recognizer on each test batch $\mathbb{B}$ to reduce character- and word-level error rates (CER/WER) under distribution shift.

**Motivation.** Domain shift is pervasive in HTR: writer idiosyncrasies, historical substrates, acquisition artifacts, layout variation, and language changes all cause $p(X, s)$ to drift from the training distribution, leading state-of-the-art supervised recognizers to exhibit sharp error spikes (Fig.1). The recent foundation model TrOCR(Li et al., 2023) shows stronger OOD generalization, but remains sub-optimal when the target distribution is not directly encountered at test time. This raises a practi-

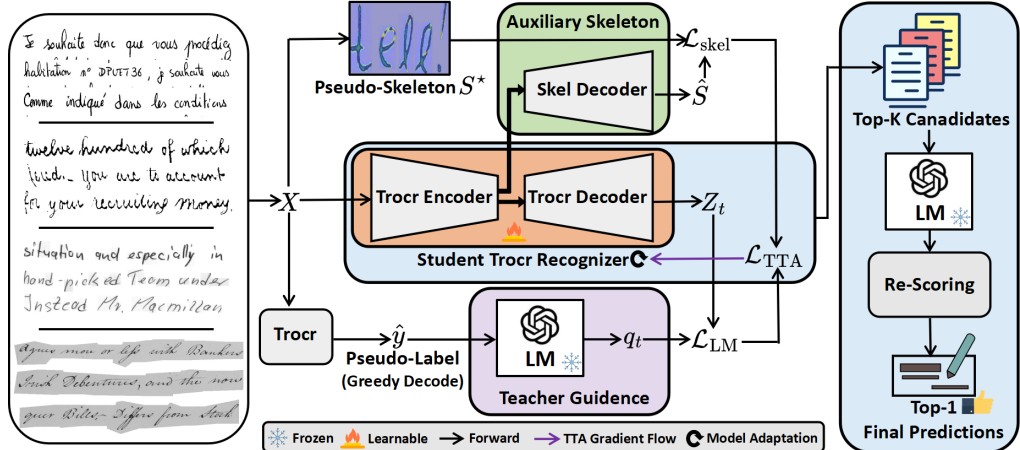

Figure 2: **Overview of our test-time adaptation (TTA) framework for HTR. Student recognizer:** a TrOCR encoder–decoder produces token logits $Z_t$ for the pseudo-label $\hat{y}$. **Teacher guidance:** a frozen GPT-2 LM returns log-probabilities for the top-$K_{\text{tok}}$ candidates, forming a fused teacher distribution $q_t$ that supervises the student via the distillation loss $\mathcal{L}_{\text{LM}}$. **Auxiliary skeleton decoder:** a stroke head predicts a soft skeleton $\hat{S}$ from encoder features and is trained against an image-derived target $S^\star$ with the geometric loss $\mathcal{L}_{\text{skel}}$. **LM reranking:** after adaptation, beam candidates are rescored with GPT-2 under sequence-level shallow fusion, and the top-1 hypothesis is selected.

cal question: *how can accuracy be recovered under deployment shift when neither source data nor target labels are available?*

Prior adaptation strategies typically rely *either* on visual heuristics, which risk linguistic drift, *or* on language-only rescoring, which ignores stroke geometry. Such decoupled updates lead to uncoordinated adaptation and limited transfer, and often require offline training on source data, restricting scalability. Motivated by these gaps, we propose a plug-and-play, *source-free* Fully-TTA framework that couples a stroke-preserving visual objective with a document-conditioned language prior, integrated into the ViT-style encoder and autoregressive decoder of TrOCR. At a high level, our method combines two lightweight priors, stroke topology from the image and document-level language regularity, within short episodic updates, then decodes once and resets. This design enables appearance cues and linguistic regularities to jointly guide on-the-fly adaptation, without source replay or supervision. As illustrated in Fig. 1, the framework stabilizes recognition under shift, consistently reduces CER/WER across heterogeneous targets, and provides a compute-bounded, practical recipe for robust HTR in diverse deployment scenarios.

## 4 METHOD

**Overview.** Our method illustrated in Fig. 2, addresses distribution shift by adapting the foundation model TrOCR on-the-fly to unlabeled target data, without any offline training. We instantiate the three design principles: **(i) Geometry preservation** via a Dice+Chamfer skeleton loss on an auxiliary stroke head; **(ii) Semantic anchoring** via an uncertainty-gated KL regularizer to a document-conditioned LM prior; and **(iii) Decoding-time regularization** via LM rescoring of an $n$-best list sequence-level shallow fusion (Sec. 4.1). We then detail the episodic test-time adaptation protocol (Sec. 4.2).

### 4.1 MODEL ARCHITECTURE

**Skeleton-aware self-supervision.** To counter the topology-breaking drift in the *vision* pathway, we impose a label-free, stroke-level self-supervision signal that preserves geometry at test-time. The vision encoder produces token features which we reshape to a spatial feature map $\boldsymbol{F} \in \mathbb{R}^{C \times H \times W}$ (we ignore the CLS token). We propose an auxiliary *stroke head* which applies a sequence of ConvTranspose2d layers with ReLU activations, progressively increasing spatial resolution while reducing channel dimensionality. A final sigmoid squashes the output into $[0, 1]$, yielding a soft skeleton map $\hat{\boldsymbol{S}} \in [0, 1]^{H' \times W'}$. Rather than reproducing pixels, the randomized stroke head leverages its architectural bias to favor coherent, self-similar centerlines, so $\hat{\boldsymbol{S}}$ emphasizes topology over exact appearance (Ulyanov et al., 2018). This auxiliary branch acts as a decoder dedicated to stroke

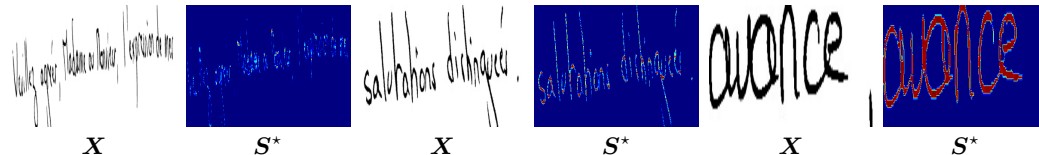

$$X \qquad S^\star \qquad X \qquad S^\star \qquad X \qquad S^\star$$

Figure 3: **Visualization of Skeleton-guided self-supervision:** The figure presents qualitative pairs $(\boldsymbol{X}, \boldsymbol{S}^\star)$. For each input line image $\boldsymbol{X}$, the target skeleton $\boldsymbol{S}^\star$ is computed in a label-free manner by inverting $\boldsymbol{X}$ and applying a single morphological erosion, yielding a thin stroke map that serves as the supervisory signal for the skeleton head.

structure, separate from the text decoder. A target skeleton $\boldsymbol{S}^\star \in [0,1]^{H' \times W'}$ is computed directly from the input by inverting the line image, applying a single morphological erosion, and resizing to $(H', W')$ to match the stroke head output. $\boldsymbol{S}^\star$ preserves stroke topology while discarding nuisance factors (width/contrast), so matching $\hat{\boldsymbol{S}}$ to $\boldsymbol{S}^\star$ supplies a label-free geometric prior for self-supervision under source-free TTA. We optimize a shape-aware loss that combines Dice and Chamfer terms:

$$\mathcal{L}_{\text{skel}}(\hat{\boldsymbol{S}}, \boldsymbol{S}^\star) = 1 - \underbrace{\frac{2\langle \hat{\boldsymbol{S}}, \boldsymbol{S}^\star \rangle + \epsilon}{\|\hat{\boldsymbol{S}}\|_1 + \|\boldsymbol{S}^\star\|_1 + \epsilon}}_{\text{Dice } (Milletariet\ al.,\ 2016)} + \gamma \underbrace{\text{Chamfer}\big(\hat{\boldsymbol{S}} > \tau_{\text{skel}},\ \boldsymbol{S}^\star > \tau_{\text{skel}}\big)}_{\text{bidirectional set distance } (Borgefors,\ 1986)}, \qquad (1)$$

with small $\epsilon > 0$, binarization threshold $\tau_{\text{skel}} \in (0,1)$, and weight $\gamma > 0$ (we use $\gamma{=}0.1$). This objective encourages stroke connectivity (Dice) and geometric alignment of centerlines (Chamfer), which are known to be robust to stroke width and contrast shifts common in HTR.

Our skeleton loss supplies a label-free *geometric prior* that anchors the vision encoder to stroke *centerlines* rather than raw intensities, making test-time updates robust under domain shift. As illustrated in Fig. 3, the input line $\boldsymbol{X}$ (left) is mapped to a target skeleton $\boldsymbol{S}^\star$ (middle) by inverting and lightly eroding $\boldsymbol{X}$, which preserves connectivity and topology while discarding nuisance factors such as width and contrast. The right panel shows an *overlay* $O(\hat{\boldsymbol{S}}, \boldsymbol{S}^\star)$: we threshold both maps at $\tau_{\text{skel}}$, upsample them to the image resolution, extract 1-px centerline contours, and superimpose those contours on the original line image $\boldsymbol{X}$; the traces lie exactly along the handwriting strokes (not the background), so coincident traces indicate agreement between $\hat{\boldsymbol{S}}$ and $\boldsymbol{S}^\star$, while offsets or missing segments indicate disagreement. Optimizing the Dice+Chamfer objective $\mathcal{L}_{\text{skel}}(\hat{\boldsymbol{S}}, \boldsymbol{S}^\star)$ drives $\hat{\boldsymbol{S}}$ to match $\boldsymbol{S}^\star$ in both coverage (connectivity) and geometry (centerline alignment), suppressing topology breaks and drift during unlabeled adaptation.

**LM-guided KL regularization (LM-KL).** To mitigate the *token-level* instability under distribution shift, we softly align the decoder's distribution to a document-conditioned LM prior through an uncertainty-gated KL term. Let $\boldsymbol{Z} \in \mathbb{R}^{T \times V}$ be the TrOCR decoder logits under teacher forcing on the pseudo-label $\hat{\boldsymbol{y}}$ (from greedy decoding, no labels), where $T$ is the target sequence length and $V$ is the tokenizer's vocabulary size. At each step $t$, we first take the top-$K$ candidate token ids $\mathbb{K}_t \subset \{1, \ldots, V\}$ from $\boldsymbol{Z}_t$ (after masking padded/special symbols). For each $k \in \mathbb{K}_t$, we query the frozen autoregressive LM with the *document context* ctx (last $M$ accepted lines) and the prefix $\hat{\boldsymbol{y}}_{<t}$ to obtain $\log p_{\text{LM}}(k \mid \hat{\boldsymbol{y}}_{<t}, \text{ctx})$. We then build a fused teacher $q_t$ supported on $\mathbb{K}_t$ via log-linear interpolation of model and LM scores (Gulcehre et al., 2015; Sriram et al., 2017):

$$q_t(k) \propto \exp\left(\tfrac{1}{\tau} z_{t,k} + \alpha \log p_{\text{LM}}(k \mid \hat{\boldsymbol{y}}_{<t}, \text{ctx})\right) \text{ if } k \in \mathbb{K}_t, \quad q_t(k) = 0 \text{ otherwise}, \qquad (2)$$

with temperature $\tau > 0$ and LM prior weight $\alpha \geq 0$. The distillation loss is defined as:

$$\mathcal{L}_{\text{LM}} = \tau^2 \sum_{t=1}^{T} \mathbf{1}_{H(p_t) > \delta}\, D_{\text{KL}}\big(q_t \,\|\, \pi_t^{(\tau)}\big), \qquad p_t = \text{softmax}(\boldsymbol{Z}_t),\ \ \pi_t^{(\tau)} = \text{softmax}(\boldsymbol{Z}_t/\tau). \qquad (3)$$

$$D_{\text{KL}}\big(q_t \,\|\, \pi_t^{(\tau)}\big) = \text{CE}\big(q_t, \pi_t^{(\tau)}\big) - H(q_t), \qquad H(p) = -\sum_k p(k) \log p(k). \qquad (4)$$

Intuitively, this KL term encourages the model distribution $\text{softmax}(\boldsymbol{Z}_t/\tau)$ to stay close to the fused teacher $q_t$, so that even without labels the decoder inherits both its own logits $z_{t,k}$ and the LM prior $\log p_{\text{LM}}(k \mid \hat{\boldsymbol{y}}_{<t}, \text{ctx})$. where the $\tau^2$ factor follows distillation practice (Hinton et al., 2015) and the entropy gate mirrors confidence-aware TTA updates, allowing the model to only

adapt when it is uncertain (Wang et al., 2020). In practice, we use fixed hyperparameters for top-$K$, temperature $\tau$, and fusion weight $\alpha$, and apply an uncertainty gate on token-level entropy $H(p_t)$ with padded/special positions masked. Since equation 4 is constant with respect to model parameters, we implement the KL divergence using the equivalent cross-entropy form.

**LM rescoring (n-best).** As the third pillar of our design, after adaptation we run beam search to produce candidates $\{\hat{\boldsymbol{y}}^{(b)}\}_{b=1}^{B}$ with model scores $\log p_{\boldsymbol{\theta}'}(\hat{\boldsymbol{y}}^{(b)} \mid \boldsymbol{X})$. We compute a length-normalized LM score $\mathcal{R}(\hat{\boldsymbol{y}}^{(b)})$ *without additional document context* and apply sequence-level shallow fusion:

$$\mathcal{R}(\hat{\boldsymbol{y}}^{(b)}) \;=\; \log p_{\boldsymbol{\theta}'}(\hat{\boldsymbol{y}}^{(b)} \mid \boldsymbol{X}) \;+\; \beta \, \frac{1}{|\hat{\boldsymbol{y}}^{(b)}|} \sum_{t=1}^{|\hat{\boldsymbol{y}}^{(b)}|} \log p_{\mathrm{LM}}(\hat{y}_t^{(b)} \mid \hat{\boldsymbol{y}}_{<t}^{(b)}). \tag{5}$$

We then select the best candidate $\arg\max_b \mathcal{R}(\hat{\boldsymbol{y}}^{(b)})$.

Our sequence-level shallow-fusion reranking follows established LM–decoder integration strategies, shallow/deep fusion in neural machine translation, and cold fusion in speech recognition, which we adapt to handwritten text recognition (Gulcehre et al., 2015; Sriram et al., 2017). It also complements classic HTR decoding frameworks that combine recognizers with external LMs (e.g., Word Beam Search, PyLaia) and prior rescoring methods for text-line recognition (Scheidl et al.,

---

**Algorithm 1** Episodic Test-Time Adaptation with Skeleton and LM Priors

**Require:** Pretrained recognizer parameters $\boldsymbol{\theta}$; test batch $\mathbb{B} = \{\boldsymbol{X}_i\}_{i=1}^{N}$ (size $N$);
**Require:** beam size $B$; update steps $U$; learning rate $\eta$; hyperparams $\tau_{\mathrm{skel}}, \delta; K$; context window $M$.
1: $\mathrm{ctx} \leftarrow [\,]$ ▷ last $M$ accepted lines per document
2: **for** each test batch $\mathbb{B} = \{\boldsymbol{X}_i\}_{i=1}^{N}$ **do**
3:    Initialize $\boldsymbol{\theta}' \leftarrow \mathrm{clone}(\boldsymbol{\theta})$
4:    **for** $u = 1, \ldots, U$ **do** ▷ few inner-loop updates
5:       Compute $\mathcal{L}_{\mathrm{skel}}(\hat{\boldsymbol{S}}, \boldsymbol{S}^{\star})$ from $\boldsymbol{X}_i \in \mathbb{B}$
6:       Compute $\mathcal{L}_{\mathrm{LM}}$ using logits $\boldsymbol{Z}$, pseudo-labels $\hat{\boldsymbol{y}}$, and context ctx
7:       $\mathcal{L}_{\mathrm{TTA}} \leftarrow \lambda_{\mathrm{skel}}\mathcal{L}_{\mathrm{skel}} + \lambda_{\mathrm{LM}}\mathcal{L}_{\mathrm{LM}}$
8:       $\boldsymbol{\theta}' \leftarrow \boldsymbol{\theta}' - \eta\nabla_{\boldsymbol{\theta}'}\mathcal{L}_{\mathrm{TTA}}$
9:    **end for**
10:   **for** $i = 1, \ldots, N$ **do** ▷ per line in the batch
11:      Beam decode $\boldsymbol{X}_i$: $\{\hat{\boldsymbol{y}}^{(b)}\}_{b=1}^{B}$
12:      $\hat{\boldsymbol{y}}_i^{\star} \leftarrow \arg\max_b \mathcal{R}(\hat{\boldsymbol{y}}^{(b)})$ ▷ LM shallow fusion
13:      Append $\hat{\boldsymbol{y}}_i^{\star}$ to ctx (keep last $M$ lines)
14:   **end for**
15:   Discard $\boldsymbol{\theta}'$ ▷ no cross-batch accumulation
16: **end for**

---

2018; Tarride et al., 2024; Wick et al., 2022). The selected line is appended to ctx (bounded to the $M$ most recent lines) to update the document-level prior for subsequent decoding without ground-truth labels.

## 4.2 INFERENCE: EPISODIC TEST-TIME ADAPTATION

At deployment, we follow an *episodic* per-instance protocol: for each test mini-batch $\mathbb{B} = \{\boldsymbol{X}_i\}_{i=1}^{N}$, we clone the pretrained recognizer $\boldsymbol{\theta}$ to a learner $\boldsymbol{\theta}'$, take a small number $U$ of gradient updates using only self-supervised signals, decode once, and then discard $\boldsymbol{\theta}'$. This bounds compute per batch and prevents cross-document drift. The adaptation loss couples a visual prior with a linguistic prior:

$$\mathcal{L}_{\mathrm{TTA}} \;=\; \lambda_{\mathrm{skel}}\,\mathcal{L}_{\mathrm{skel}} + \lambda_{\mathrm{LM}}\,\mathcal{L}_{\mathrm{LM}}, \tag{6}$$

where $\mathcal{L}_{\mathrm{skel}}$ preserves stroke topology/connectivity and $\mathcal{L}_{\mathrm{LM}}$ distills token distributions toward a document-aware LM teacher; updates follow $\boldsymbol{\theta}' \leftarrow \boldsymbol{\theta}' - \eta\nabla_{\boldsymbol{\theta}'}\mathcal{L}_{\mathrm{TTA}}$. After adaptation, each line is decoded with beam search and LM reranking, and the selected hypothesis $\hat{\boldsymbol{y}}^{\star}$ is appended to a context buffer ctx (keeping the last $M$ lines) to sharpen future LM priors without leaking ground-truth labels.

**Summary.** Algorithm 1 outlines our episodic procedure. Each batch is adapted independently to prevent drift, while visual and linguistic priors jointly stabilize recognition under distribution shift. The framework is fully *source-free* and label-free: skeleton guidance provides a spatial, content-agnostic signal tied to handwriting geometry, and LM fusion with reranking injects corpus-level priors aligned with the target language and domain (Scheidl et al., 2018; Tarride et al., 2024). Together with episodic updates (Sun et al., 2020; Wang et al., 2020), this yields a simple yet effective TTA recipe for robust HTR under deployment shift.

**Implementation details.** All experiments are run on a single NVIDIA RTX,4090 GPU. We operate on line crops provided by each dataset; images are converted to a TrOCR-compatible format (grayscale/RGB as required), padded to preserve aspect ratio, and resized to the encoder's native resolution. For stability and reproducibility, we filter special tokens from fusion supports to prevent degenerate targets, gate adaptation updates by confidence to avoid over-fitting on easy steps,

and disable LM contributions whenever any non-finite scores are detected. A single configuration is used across datasets (fixed inner-loop budget, decoding settings, fusion temperatures/weights); the external LM remains frozen, we avoid dataset-specific tuning, and we rely on public checkpoints/preprocessing. Utility code (e.g., resume hooks, context buffers) does not affect the evaluation protocol. In all experiments, we fix the context buffer to $M = 20$ lines—a small window (roughly 1–2 paragraphs) that we found sufficient to capture local lexical patterns without incurring large prompts or memory overhead, and we use the same value across IAM, GW, Bentham, and RIMES rather than tuning $M$ per dataset. *See Appendix A for more details.*

## 5 EXPERIMENTS

### 5.1 DATASETS AND SETUP

**Benchmarks and evaluation protocols.** We evaluate on four handwriting corpora spanning historical/modern and cross-language settings: **GW**—18th-century English letters with highly cursive hands and degraded paper (Rath et al., 2004); **IAM**—modern English from 657 writers (Marti & Bunke, 2002); **RIMES**—French mail, serving as a *cross-language* target (Kermorvant & Louradour, 2010); and **Bentham**—19th-century English manuscripts (Causer & Wallace, 2012). Together, they probe robustness to writer variation, historical vs. modern appearance, and English,$\rightarrow$, French shift. *Splits.* We use the *Aachen* split for **IAM** and the official splits for **GW**, **RIMES, and Bentham**. We use a learning rate of $1 \times 10^{-4}$ and a mini-batch size of 8 for all experiments. We report CER and WER with standard normalization, and edit distances follow the code used for evaluation.

**Backbone recognizer.** While many VLM/LLM backbones exist, almost none are tailored to handwriting; hence, we adopt *TrOCR* (Li et al., 2023), a handwriting-specific foundation model built as a Transformer encoder–decoder with a ViT visual encoder and an autoregressive text decoder, pretrained on large handwriting corpora. As shown in Table 5, even strong recent generic OCR/VLM systems (Donut, GOT-OCR2, Qwen3-VL-4B) still yield much higher CER/WER on IAM than TrOCR, indicating that current MLLMs/VLMs do not yet replace a dedicated HTR backbone(Kim et al., 2022; Wei et al., 2024; Yang et al., 2025). We therefore use TrOCR as a strong and realistic base recognizer for studying test-time adaptation. In our *source-free* setting, we evaluate four public checkpoints as the student recognizer: **TrOCR-base-stage1** (pretraining only; no fine-tuning), and the fine-tuned **TrOCR-small**, **TrOCR-base**, and **TrOCR-large**. For the main comparisons, we use **stage1** on **IAM** to emulate a truly OOD scenario, and **base** on **RIMES**, **GW**, and **Bentham**; we also report results for **small** and **large** to assess backbone sensitivity (Table 1). No target-domain labels or source training data are used at deployment, and during adaptation we update all model parameters (encoder and decoder).

**Language model (teacher).** We use a *frozen* **GPT-2 Large** to guide test-time adaptation in two roles aligned with our method: (i) **LM-guided KL regularization (LM-KL)**—scoring the top-$K_{tok}$ tokens conditioned on the rolling document context `ctx` to form $q_t$; and (ii) **LM rescoring (n-best)**—returning length-normalized log-likelihoods for complete candidates *without* additional document context. We choose GPT-2 Large because it supplies calibrated, left-to-right *per-token* log-probabilities required by our KL regularizer and shallow-fusion rescoring; its byte-level BPE(Byte-Pair Encoding) can score any string—including punctuation, diacritics, and mixed case—without losing information or introducing unknown tokens; and at $\sim$774M parameters, it fits on a single GPU in FP16 for low-latency scoring. In contrast, *PyLaia* integrates $n$-gram LMs for CTC decoding rather than a causal Transformer aligned with our seq2seq decoder (Tarride et al., 2024). For completeness, we also evaluate alternative teachers (e.g., distilgpt2 and CamemBERT) to demonstrate robustness across LMs (Table 4).

### 5.2 COMPARISON WITH ~~SOTA~~ SPECIALIZED MODELS EVALUATED ON OOD DATA

Table 1 reports our gains when integrated into four TrOCR backbones—**stage-1** (pretrained only; no finetuning), **small**, **base**, and **large**. We use **TrOCR-base-stage1** as the backbone for all main OOD experiments on IAM. Summarizing by averages across the four datasets (IAM/RIMES/GW/Bentham): on **stage-1**—ideal for truly OOD evaluation on IAM, our method reduces CER by **2.96** points on average (**16.7%** rel.) and WER by **7.38** points (**18.6%** rel.). For **TrOCR-small**, the average drops are **3.17** CER (**17.3%**) and **2.24** WER (**7.8%**); for **TrOCR-base**, **4.36** CER (**27.1%**) and **4.10** WER (**12.6%**); and for **TrOCR-large**, **3.16** CER (**21.9%**) and **3.36** WER (**12.1%**). Overall, averaging across all backbones and datasets, our approach yields a mean

Table 1: **Comparison between our method and prior SOTAspecialized models evaluated on OOD data, the error rates (lower is better).** Legacy baselines (CRNN–VLT) are reproduced from (Garrido-Munoz & Calvo-Zaragoza, 2025) published results. For TrOCR, we report small, base-stage1, base, and large models, and show improvements with our method. Best results are shown in **bold**.

| Model | IAM | | Rimes | | GW | | Bentham | |
|---|---|---|---|---|---|---|---|---|
| | CER↓ | WER↓ | CER↓ | WER↓ | CER↓ | WER↓ | CER↓ | WER↓ |
| CRNN (Puigcerver, 2017) | 34.9 | 68.2 | 25.0 | 74.2 | 31.1 | 68.8 | 25.3 | 54.9 |
| VAN (Coquenet et al., 2022) | 28.6 | 76.8 | 21.3 | 66.9 | 32.0 | 74.7 | 26.6 | 62.1 |
| C-SAN (d'Arce et al., 2022) | 31.5 | 83.6 | 29.8 | 80.7 | 49.8 | 93.9 | 38.9 | 82.2 |
| HTR-VT (Li et al., 2025b) | 33.7 | 83.7 | 28.3 | 78.7 | 38.6 | 83.3 | 33.3 | 70.8 |
| Kang (Kang et al., 2022) | 42.1 | 87.1 | 32.0 | 78.3 | 44.0 | 77.6 | 39.4 | 68.2 |
| Michael (Michael et al., 2019) | 49.1 | 82.9 | 35.5 | 84.5 | 43.6 | 76.1 | 43.5 | 69.6 |
| LT (Barrere et al., 2022) | 42.0 | 72.0 | 30.8 | 77.4 | 32.3 | 65.7 | 33.8 | 60.6 |
| VLT (Barrere et al., 2024) | 41.3 | 70.3 | 29.4 | 76.1 | 32.1 | 65.8 | 33.3 | 60.5 |
| TrOCR-stage1 (Li et al., 2023) | 11.78 | 35.86 | 23.32 | 55.30 | 14.61 | 31.66 | 31.77 | 49.63 |
| **Our Method** | **8.06** | **17.65** | **17.93** | **48.39** | **13.63** | **29.79** | **30.02** | **47.09** |
| TrOCR-small (Li et al., 2023) | 7.13 | 15.49 | 28.99 | 67.64 | 22.35 | 36.22 | 20.55 | 31.69 |
| **Our Method** | **5.63** | **13.32** | **25.80** | **65.57** | **18.24** | **34.17** | **16.69** | **29.03** |
| TrOCR-base (Li et al., 2023) | 5.10 | 11.95 | 27.54 | 65.25 | 16.86 | 29.07 | 17.55 | 28.02 |
| **Our Method** | **3.61** | **10.88** | **21.29** | **59.04** | **12.40** | **25.40** | **12.31** | **22.59** |
| TrOCR-large (Li et al., 2023) | 3.71 | 9.58 | 23.94 | 59.82 | 15.18 | 26.21 | 11.76 | 19.99 |
| **Our Method** | **3.17** | **8.78** | **18.75** | **54.58** | **11.18** | **21.56** | **8.84** | **17.25** |

Table 2: **Test-time adaptation (TTA) on four handwriting benchmarks.** We report CER/WER (lower is better) for *TrOCR-stage1* and *TrOCR-base* backbones under several TTA strategies: vanilla BatchNorm adaptation (BN), TENT (entropy minimization), and EATA. Our method consistently achieves the lowest error on IAM, Rimes, GW, and Bentham; best results are in **bold**. The last column (**Speed**) reports runtime in milliseconds per sample (ms/sample), averaged over 100 test lines. For a deployment-friendly and parameter-efficient setting comparable to TENT/EATA, we also report a lighter variant ("Our Method LN/bias-update-only") using the same GPT-2-Large teacher, which updates only LayerNorm weight/bias in the TrOCR backbone.

| TTA Methods | IAM | | Rimes | | GW | | Bentham | | Speed |
|---|---|---|---|---|---|---|---|---|---|
| | CER↓ | WER↓ | CER↓ | WER↓ | CER↓ | WER↓ | CER↓ | WER↓ | ms/sample |
| TrOCR-stage1 (Li et al., 2023) | 11.78 | 35.86 | 23.32 | 55.30 | 14.61 | 31.66 | 31.77 | 49.63 | 103.54 |
| BN-stage1 (Ioffe & Szegedy, 2015) | 101.09 | 102.58 | 112.79 | 124.55 | 152.02 | 175.31 | 115.45 | 102.54 | +36.53 |
| TENT-stage1 (Wang et al., 2020) | 11.19 | 35.27 | 19.69 | 51.09 | 13.95 | 30.41 | 31.98 | 48.64 | +55.45 |
| EATA-stage1 (Niu et al., 2022) | 11.71 | 35.94 | 23.29 | 54.95 | 14.88 | 32.38 | 30.98 | 48.45 | +76.07 |
| **Our Method** | **8.06** | **17.65** | **17.93** | **48.39** | **13.63** | **29.79** | **30.02** | **47.09** | +313.46 |
| TrOCR-base (Li et al., 2023) | 5.10 | 11.95 | 27.54 | 65.25 | 16.86 | 29.07 | 17.55 | 28.02 | 103.54 |
| BN-base (Ioffe & Szegedy, 2015) | 104.84 | 149.83 | 109.78 | 160.76 | 227.19 | 143.65 | 102.06 | 186.33 | +36.53 |
| TENT-base (Wang et al., 2020) | 4.77 | 11.32 | 23.92 | 61.44 | 16.28 | 27.64 | 15.71 | 25.47 | +55.45 |
| EATA-base (Niu et al., 2022) | 4.76 | 11.45 | 26.26 | 62.43 | 15.61 | 26.94 | 14.99 | 24.97 | +76.07 |
| Our Method LN/bias-update-only | 3.68 | 10.94 | 21.46 | 59.53 | 12.91 | 25.84 | 12.94 | 23.30 | +62.93 |
| **Our Method** | **3.61** | **10.88** | **21.29** | **59.04** | **12.40** | **25.40** | **12.31** | **22.59** | +313.46 |

absolute reduction of **3.41** CER and **4.27** WER, achieved under *strict source-free* TTA by coupling a stroke-preserving visual objective with a lightweight, document-aware language prior.

**Positioning Against Established HTR Systems** The upper block of Table 1 reports reference results for representative HTR systems; these numbers are *reproduced from* Garrido-Munoz & Calvo-Zaragoza (2025), and for each method we quote the *best-performing source* among those compiled in that survey. Under our evaluation protocol, their CER intervals are approximately 28.6–49.1 on **IAM**, 21.3–35.5 on **RIMES**, 31.1–49.8 on **G.W.**, and 25.3–43.5 on **Bentham**. Even the TrOCR *baseline* already lies below these intervals (e.g., 11.78 CER on IAM), reflecting the strength of modern encoder–decoder backbones. Our test-time adaptation further reduces error to **8.06/17.65** (CER/WER) on IAM—measured with the **TrOCR-stage1** (pretrained-only) backbone—and to **21.29/59.04** on RIMES, **12.40/25.40** on G.W., and **12.31/22.59** on Bentham using the **TrOCR-base** backbone. Because training corpora and decoding setups vary across historical sys-

Table 3: **Ablation of the proposed components across four datasets.** Lower **CER/WER** shows better results.

| Ablation Study | IAM CER↓ | IAM WER↓ | RIMES CER↓ | RIMES WER↓ | GW CER↓ | GW WER↓ | Bentham CER↓ | Bentham WER↓ |
|---|---|---|---|---|---|---|---|---|
| Frozen+LM-only | 10.08 | 21.06 | 21.80 | 60.84 | 13.51 | 26.83 | 13.49 | 24.50 |
| w/o skeleton | 9.33 | 20.82 | 21.42 | 59.43 | 12.86 | 25.79 | 12.47 | 22.91 |
| w/o LM-Fusion KL | 9.40 | 20.94 | 21.71 | 60.29 | 13.91 | 26.92 | 12.95 | 23.65 |
| w/o LM reranking | 10.44 | 24.08 | 21.87 | 61.61 | 14.06 | 28.35 | 14.19 | 26.42 |
| **Our method** | **8.06** | **17.65** | **21.29** | **59.04** | **12.40** | **25.40** | **12.31** | **22.59** |

Table 4: **External LM variants.** [†]RIMES uses Camembert-base; others uses distilgpt2. The gpt2-large is our choice.

| LM (teacher) | IAM CER↓ | IAM WER↓ | Rimes CER↓ | Rimes WER↓ | GW CER↓ | GW WER↓ | Bentham CER↓ | Bentham WER↓ |
|---|---|---|---|---|---|---|---|---|
| Baseline (no LM) | 11.78 | 35.86 | 27.54 | 65.25 | 16.86 | 29.07 | 17.55 | 28.02 |
| distilgpt2/Camembert[†] | 9.67 | 22.63 | **20.72** | 59.10 | 13.63 | 27.10 | 13.83 | 25.29 |
| gpt2-large | **8.06** | **17.65** | 21.29 | **59.04** | **12.40** | **25.40** | **12.31** | **22.59** |

Table 5: **Comparison of newest and popular generic OCR/VLM baselines on IAM.** We report CER/WER (lower is better) on the IAM test set. Despite rapid progress in general-purpose OCR/VLM models, they still underperform a specialized TrOCR backbone on line-level HTR.

| | TrOCR-base (Li et al., 2023) CER↓ | WER↓ | Donut (Kim et al., 2022) CER↓ | WER↓ | GOT-OCR2 (Wei et al., 2024) CER↓ | WER↓ | Qwen3-VL-4B (Yang et al., 2025) CER↓ | WER↓ |
|---|---|---|---|---|---|---|---|---|
| IAM | **5.10** | **11.95** | 28.00 | 44.00 | 60.76 | 80.24 | 7.68 | 31.73 |

Table 6: **TTA on fine-tuned models (GW, RIMES).** We compare a fine-tuned TrOCR-base *without* TTA to the same model *with* our test-time adaptation. We report CER/WER (lower is better). Our method further reduces error post fine-tuning while using no source data at test time; best results are in **bold**.

| TTA Methods | GW CER↓ | GW WER↓ | RIMES CER↓ | RIMES WER↓ | Bentham CER↓ | Bentham WER↓ |
|---|---|---|---|---|---|---|
| Fine-tuned (no TTA) | 7.92 | 16.10 | 22.33 | 47.73 | 16.80 | 26.78 |
| **Fine-tuned + Our Method** | **7.53** | **14.76** | **14.09** | **39.46** | **12.65** | **23.50** |

tems, we treat these figures as contextual reference points rather than strict SOTA claims; within this context, our results establish strong deployment-time performance under source-free constraints.

**TTA Strategy Comparison Across TrOCR Backbones.** We compare three TTA baselines (BN, TENT, EATA) of Table 2 on **TrOCR-stage1** (pretrained only) and **TrOCR-base**. BN adaptation is unstable and often degrades performance, whereas TENT and EATA yield modest gains over the frozen baseline. Our method is consistently strongest: on **TrOCR-base** it lowers CER by **1.49/6.25/4.46/5.24** on IAM/RIMES/GW/Bentham (avg $-4.36$, $\approx 27\%$ relative) and reduces WER by **1.07/6.21/3.67/5.43** (avg $-4.10$). On **TrOCR-stage1**—a truly OOD setting for IAM—our method attains **8.06/17.65** (CER/WER) on IAM and achieves the lowest *CER* on RIMES, GW, and Bentham within that backbone; WER trends for stage-1 are mixed, reflecting decoding without dataset-specific tuning. Overall, our approach delivers the most reliable and largest gains across datasets and backbones. (See Appendix C for implementation details.)

**Latency–accuracy trade-off and deployable variants.** On IAM, our baseline TrOCR decoding (no TTA, no skeleton, no LM) runs at **103.54 ms/line**, while the *full* pipeline with GPT-2-Large and all encoder+decoder parameters updated adds **+11.00 ms/line** for skeleton TTA, **+288.49 ms/line** for LM-Fusion KL, and **+14.23 ms/line** for LM reranking (117.77 total $-$ 103.54 baseline), for a total of $\approx$**417 ms/line** ($\approx$**4.0**$\times$ baseline), with the overhead dominated by LM-Fusion KL and skeleton+reranking themselves adding only about +25% on top of baseline decoding. To make our TTA setting closer to realistic deployment and directly comparable to parameter-efficient baselines such as TENT/EATA, we also evaluate a much lighter *LN/bias-only* variant with the same GPT-2-Large teacher, which $< 0.2\%$ of the TrOCR-base backbone ($\approx 4 \times 10^5$ of $3.34 \times 10^8$ parameters) and costs **7.98 ms/line** for skeleton TTA, **42.27 ms/line** for LM-Fusion KL, and **116.22 ms/line** for LM reranking, i.e., $\approx$**166.47 ms/line** ($\approx$**1.6**$\times$ baseline), while achieving and recovering $\approx$88–97% of the full GPT-2-Large gains shown in Table 2; if needed, one can further trade accuracy for speed by using a smaller LM (e.g., DistilGPT-2, $\approx$**1.8**$\times$ baseline), disabling LM-Fusion, or reducing the number of adaptation steps $U$, making the method tunable to different latency budgets rather than tied to a single heavy configuration.

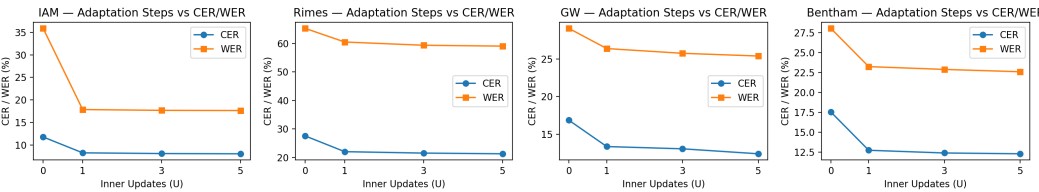

Figure 4: **Effect of test-time adaptation on four HTR benchmarks.** Each panel plots CER (circles) and WER (squares), in %, versus the number of inner updates $U \in \{0, 1, 3, 5\}$ (lower is better).

## 5.3 ABLATION STUDIES

**Ablations: Skeleton, LM-KL, and LM Reranking.** Table 3 quantifies the contribution of each component by removing it from the full system. Three patterns are consistent. (i) *Skeleton guidance* matters across all datasets, most notably on IAM: removing it degrades IAM from 8.06/17.65 to 9.33/20.82 (CER/WER; +1.27/+3.17). Historical sets also benefit (GW: 12.40/25.40 → 12.86/25.79; +0.46/+0.39), indicating that stroke-level cues regularize the encoder under writer/style variation. (ii) *LM-Fusion KL* (token-level teacher) stabilizes lexical choices, with clear gains on language-shifted or visually noisy data: removing it hurts RIMES (21.29/59.04 → 21.71/60.29; +0.42/+1.25) and Bentham (12.31/22.59 → 12.95/23.65; +0.64/+1.06), and also impacts IAM (8.06/17.65 → 9.40/20.94; +1.34/+3.29) and GW (12.40/25.40 → 13.91/26.92; +1.51/+1.52). (iii) *LM reranking* (sequence-level shallow fusion) is the single most impactful linguistic component: removing it consistently yields the largest WER regressions (IAM: +6.43 points; GW: +2.95; Bentham: +3.83; RIMES: +2.57), underscoring the value of sequence-level LM scoring for pruning implausible hypotheses.

**Language-Model Choice for Decoding and Reranking.** Table 4, we ablate the choice of external language model for decoding/reranking. For the French corpus (RIMES), we use French special LM CamemBERT-base, and for the English datasets (IAM/GW/Bentham), we use distilGPT-2, as these teachers match the target language and offer a compact, latency-friendly option that preserves our test-time budget. A larger teacher yields the most reliable gains and becomes our default; crucially, this improvement is orthogonal to our adaptation losses and does not change the inner-loop budget, isolating the benefit to stronger language guidance.

**Effect of Inner-Loop Update Steps ($U$).** Fig. 4 plots CER and WER versus the number of inner adaptation steps $U \in \{0, 1, 3, 5\}$ on four HTR benchmarks. Increasing $U$ consistently reduces both metrics, with the largest gain from $U=0 \to 1$, indicating an effective yet compute-efficient update. Disproportionate WER gains reflect sequence-level constraints from LM rescoring, whereas steady CER improvements arise from stroke-aware encoder regularization. Performance plateaus around $U \approx 3$, suggesting stable, non-drifting updates rather than overfitting, and corroborating that the visual and linguistic priors are complementary and jointly necessary for reliable source-free TTA.

**Discussion.** Taken together, the ablations indicate that the three components—skeleton guidance, LM-Fusion KL, and LM reranking—are mutually reinforcing. Skeleton supervision enhances robustness to stroke variability, whereas the LM terms regularize decoding at the token and sequence levels. In concert, they deliver the lowest CER/WER across all datasets, demonstrating that a hybrid visual–linguistic adaptation is the most effective strategy for source-free TTA and handwriting behaves as a structured, sequential signal shaped jointly by visual geometry and linguistic priors. *See Appendix Fig. 5 for qualitative examples; Appendix A details reproducibility, limitations, and discussion.*

## 6 CONCLUSION

We present a *source-free* test-time adaptation (TTA) framework for HTR that jointly exploits lightweight *visual* and *linguistic* priors, requiring neither source data nor target labels. Coupling a stroke-structure loss that preserves topology with a document-conditioned language-model prior that stabilizes token distributions, our method yields consistent reductions in character- and word-error rates (CER/WER) on GW, IAM, RIMES, and Bentham with a TrOCR backbone. The results indicate robust generalization under writer variation, script/language differences, and scanning artifacts, while maintaining modest computational cost and practical deployability. These breakthroughs not only bridge a critical gap in HTR robustness but also establish a scalable, deployable paradigm for resilient recognition systems, paving the way for future HTR works.

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

# APPENDIX

## A  REPRODUCIBILITY, LIMITATIONS, AND DISCUSSION

**LLM usage.**  We use a large language model solely to polish wording and improve clarity.

**Reproducibility.**  All experiments follow publicly documented preprocessing and official dataset splits (IAM: Aachen; GW, RIMES, Bentham: official). We use released official TrOCR checkpoints and standard evaluation code for CER/WER. Hyperparameters (e.g., learning rate 1e−4, batch size 8, decoding settings, adaptation steps) and hardware are specified in the implementation section and appendix; no source-domain data or target labels are used at test time. We will release the source code upon the paper acceptance.

In all experiments, we keep the TTA hyperparameters fixed across datasets and backbones: we use a document context buffer of $N_{ctx} = 20$ previous lines, learning rate $10^{-4}$, skeleton loss weight $\lambda_{skel} = 0.2$, LM-fusion weight $\lambda_{LM} = 0.1$, fusion strength $\alpha = 0.3$, LM temperature $\tau = 2.0$, token-level top-$K = 32$ for the KL term, LM reranking weight $\beta = 0.3$, and a skeleton head that outputs $112 \times 112$ maps with a binarization threshold of $0.5$ in the Chamfer loss.

**Limitations and discussion.**  One limitation is that performance depends on the out-of-distribution generalization of the underlying foundation model. Coverage is not guaranteed across all languages or handwriting styles—especially minority or low-resource languages—so gaps in the base model can cap overall accuracy. Our method does not assume a particular architecture and can run on top of different foundation models as long as final features and logits are exposed. Consequently, pairing it with stronger or more multilingual handwriting foundation models is a straightforward path to further gains. Future work includes strengthening the language prior for minority scripts and exploring script-independent visual priors to reduce dependence on base-model coverage. Our adaptation is applied independently at the line level, so when a page contains multiple writing styles or writers, the skeleton branch and gradient updates operate only on the current line image, allowing visually different writers on the same page to be handled separately.

## B  ADDITIONAL EXPERIMENTS

**Finetuning gains.**  Starting from a strong *fine-tuned* TrOCR-base, our test-time adaptation delivers clear improvements on both domains (Table 7). On **GW**, error drops from 7.92/16.1 to **7.53/14.76** (CER/WER; $-0.39/-1.34$, $\approx$**4.9%/8.3%** relative). On **RIMES**, the gains are larger, from 22.33/47.73 to **14.09/39.46** ($-8.24/-8.27$, $\approx$**36.9%/17.3%**). These results confirm that our method complements supervised fine-tuning: it adapts the model at inference to the target distribution and yields additional accuracy without using source data or target labels.

Tables 10 and 11 evaluate our method with **OLMo2** (a 2025 LM) and **Qwen3-VL-4B** as a modern MLLM baseline on IAM. Replacing GPT-2-Large with OLMo2 in our TTA pipeline yields essentially the same CER/WER (e.g., 8.17/17.78 vs. 8.06/17.65 for TrOCR-stage1, and 3.67/10.72 vs. 3.61/10.88 for TrOCR-base), indicating that our approach is *not* tied to a specific legacy LM and remains effective with newer language models. At the same time, a specialized TrOCR-base recognizer with LM reranking (3.61/10.88) still substantially outperforms the much larger Qwen3-VL-4B MLLM, even when Qwen3-VL-4B is combined with an external LM (7.39/30.97). These results show that our stroke- and LM-regularized TTA remains effective and competitive when newer foun-

Table 7: **TTA on fine-tuned models (GW, RIMES).** We compare a fine-tuned TrOCR-base *without* TTA to the same model *with* our test-time adaptation. Entries are CER/WER (lower is better). Our method further reduces error post fine-tuning while using no source data at test time; best results are in **bold**.

| TTA Methods | GW | | RIMES | | Bentham | |
|---|---|---|---|---|---|---|
| | CER↓ | WER↓ | CER↓ | WER↓ | CER↓ | WER↓ |
| Fine-tuned (no TTA) | 7.92 | 16.1 | 22.33 | 47.73 | 16.80 | 26.78 |
| **Fine-tuned + Our Method** | **7.53** | **14.76** | **14.09** | **39.46** | 12.65 | 23.50 |

Table 8: In-distribution (ID) and out-of-distribution (OOD) results (WER %) for HTR models across datasets. The OOD result is reported from the best-performing source. Results marked with * indicate outliers, meaning that the model did not converge in the ID setting. Average results (bottom row) are computed by filtering out outliers. *All values in this table are reproduced directly from the published results of Garrido et al. (Garrido-Munoz & Calvo-Zaragoza, 2025).*

| Dataset | CRNN (Puigcerver, 2017) | | VAN (Coquenet et al., 2022) | | C-SAN (d'Arce et al., 2022) | | HTR-VT (Li et al., 2025b) | | Kang (Kang et al., 2022) | | Michael (Michael et al., 2019) | | LT (Barrere et al., 2022) | | VLT (Barrere et al., 2024) | |
|---|---|---|---|---|---|---|---|---|---|---|---|---|---|---|---|---|
| | ID | OOD | ID | OOD | ID | OOD | ID | OOD | ID | OOD | ID | OOD | ID | OOD | ID | OOD |
| IAM | 22.4 | 68.2 | 24.2 | 76.8 | 50.7 | 83.6 | 18.2 | 83.7 | 23.2 | 87.1 | 20.2 | 82.9 | 23.4 | 72.0 | 27.0 | 70.3 |
| Rimes | 11.5 | 74.2 | 18.2 | 66.9 | 45.1 | 80.7 | 24.5 | 78.7 | 14.8 | 78.3 | 20.0 | 84.5 | 13.2 | 77.4 | 13.3 | 76.1 |
| G.W. | 26.0 | 68.8 | 31.2 | 74.7 | 33.8 | 93.9 | 71.7* | 83.3 | 104.3* | 77.6 | 80.1* | 76.1 | 195.4* | 65.7 | 51.4* | 65.8 |
| Bentham | 13.2 | 54.9 | 20.5 | 62.1 | 30.2 | 82.2 | 24.6 | 70.8 | 19.8 | 68.2 | 19.8 | 69.6 | 14.3 | 60.6 | 15.5 | 60.5 |
| Average | 19.2 | 79.9 | 24.8 | 81.3 | 37.2 | 90.7 | 25.4 | 87.0 | 20.1 | 87.6 | 22.3 | 87.6 | 21.6 | 82.6 | 25.7 | 82.0 |

Table 9: Best source domain for each target (rows) across all architectures. *All values in this table are reproduced directly from the published results of Garrido et al. (Garrido-Munoz & Calvo-Zaragoza, 2025).*

| Dataset | CRNN (Puigcerver, 2017) | VAN (Coquenet et al., 2022) | C-SAN (d'Arce et al., 2022) | HTR-VT (Li et al., 2025b) | Kang (Kang et al., 2022) | Michael (Michael et al., 2019) | LT (Barrere et al., 2022) | VLT (Barrere et al., 2024) |
|---|---|---|---|---|---|---|---|---|
| IAM | Bentham | Rimes | Rimes | Rimes | Rimes | Bentham | Bentham | Bentham |
| Rimes | IAM | IAM | IAM | IAM | IAM | IAM | IAM | IAM |
| G.W. | IAM | Bentham | IAM | Bentham | IAM | IAM | IAM | IAM |
| Bentham | IAM | IAM | IAM | IAM | IAM | IAM | IAM | IAM |

dation models and LMs are considered, and that a strong HTR front-end is still needed for accurate line-level recognition.

**Qualitative Examples.** The Fig. 5 illustrates qualitative improvements from our TTA on four datasets (IAM, RIMES, GW, Bentham). Rows show the input line, the transcript *before* and *after* TTA, and the ground truth (GT). Across cases, TTA fixes diverse failure modes: on IAM it repairs a word-internal character; on RIMES it restores accents and the correct lexical form; on GW it resolves historical numerals and punctuation in a dated phrase; and on Bentham it corrects visually confusable letters that flipped word identity ("Front his preview" → "From this review"). These edits reflect complementary effects of the stroke-preserving visual objective (cleaner glyph shapes) and the document-aware LM prior (more plausible words), yielding outputs closer to the ground truth.

## C  TEST-TIME ADAPTATION VARIANTS FOR TROCR

**TENT.** We implement a TENT-style adaptation tailored to TrOCR's encoder–decoder pipeline. For each unlabeled test batch, we first decode to obtain pseudo-labels, then run a brief teacher-forced update that optimizes a self-supervised objective (token-entropy minimization or pseudo-label cross-entropy). To preserve stability, updates are restricted to a small set of parameters (LayerNorm affine terms across encoder/decoder and decoder linear biases) while the model remains in evaluation mode. We optionally gate the loss by token-wise entropy, use a small learning rate and few steps with gradient clipping, then regenerate the sequence; adaptation is applied independently per batch with no cross-batch accumulation.

| | IAM | Rimes | WG | Bentham |
|---|---|---|---|---|
| Image | assuredness "Bella Bella Marie" Charlophone, a lively song that changes mid-way. | Comme indiqué dans les conditions particulières de mon contrat d'assurance | letters Orders and Instructions December 175s | From this review then it appears that |
| BeforeTTA | assuredness " Bella Bella House " ( Parlophone ) a lively song that changes tempo mid-way | Comme indique dans les conditions particulities de man cantrat d'assurance | letters Orders and Instructions . December I5 | Front his preview then it appears that |
| AfterTTA | assuredness " Bella Bella House " ( Parlophone ) a lively song that changes tempo mid-way | Comme indiqué dans les conditions particulées de man contrat d'assurance | letters Orders and Inspections. December 175s | From this review then it appears that |
| GT | assuredness " Bella Bella Marie " ( Parlophone ) , a lively song that changes tempo mid-way . | Comme indiqué dans les conditions particulières de mon contrat d'assurance | Letters Orders and Instructions. December 1755 | From this review then it appears, that |

Figure 5: **Qualitative TTA effects across datasets.** Examples from **IAM**, **RIMES**, **GW**, and **Bentham**. Rows show the input line, transcript *before* and *after* TTA, and ground truth (GT). Green highlights indicate characters corrected by TTA; Red highlights mark remaining errors. TTA repairs glyph-level confusions (e.g., letters/accents, numerals, punctuation) and yields outputs closer to GT.

Table 10: New 2025 LM (OLMo2) with our TTA.

| New LM | IAM |
|---|---|
| TrOCR-stage1 + TTA + OLMo2 | 8.17 / 17.78 |
| TrOCR-stage1 + TTA + GPT-2-Large | 8.06 / 17.65 |
| TrOCR-base + TTA + OLMo2 | 3.67 / 10.72 |
| TrOCR-base + TTA + GPT-2-Large | 3.61 / 10.88 |

Table 11: New 2025 foundation MLLM baseline.

| New Baseline | IAM |
|---|---|
| Qwen3-VL-4B | 7.68 / 31.73 |
| Qwen3-VL-4B + LM | 7.39 / 30.97 |
| TrOCR-base | 5.10 / 11.95 |
| TrOCR-base + LM | 3.61 / 10.88 |

**EATA.** We adapt TrOCR by minimizing token entropy with an additional EWC-style penalty that keeps the model close to its pre-adaptation state. Concretely, only LayerNorm affine terms are updated and a Fisher-weighted quadratic regularizer is estimated from a few unlabeled batches. As with TENT, adaptation is brief and conservative and is applied independently per batch; after the update we decode and report predictions without using source data or target labels.

**BN.** We implement a lightweight BatchNorm-based adaptation by replacing encoder `LayerNorm` blocks with BatchNorm on the last hidden dimension, initializing affine terms from the original `LayerNorm`. At test time, all weights are frozen and the model stays in `eval` mode while BN layers are set to `train` so that only running means/variances update. We expose BN to the target distribution via a few short unlabeled decode passes, then decode normally; small batches and few adaptation mini-batches keep compute overhead minimal and the evaluation protocol unchanged.

Table 12: Notation used in Sections 3–4.

| Symbol | Type / shape | Meaning |
|---|---|---|
| $\boldsymbol{X}$ | image, $H \times W$ | Input line image (grayscale/RGB); test-time only (no labels). |
| $\boldsymbol{s} = (s_1, \ldots, s_T)$ | tokens, length $T$ | Target token sequence; $\hat{\boldsymbol{y}}$ denotes the pseudo-label/decoded sequence used for teacher forcing. |
| $V$ | scalar | Vocabulary size. |
| $\boldsymbol{\theta}, \boldsymbol{\theta}'$ | params | Pretrained TrOCR parameters; adapted (episodic) copy. |
| $p_{\boldsymbol{\theta}}(\boldsymbol{s} \mid \boldsymbol{X})$ | prob. | Recognizer distribution (TrOCR). |
| $\boldsymbol{Z} \in \mathbb{R}^{T \times V}$ | logits | Decoder logits under teacher forcing on $\hat{\boldsymbol{y}}$. |
| $z_{t,k}$ | scalar | Logit for token $k$ at time $t$. |
| $p_t = \text{softmax}(\boldsymbol{Z}_t)$ | prob. over $V$ | Model token distribution at step $t$. |
| $H(p_t)$ | scalar | Entropy $-\sum_k p_t(k) \log p_t(k)$. |
| $q_t$ | prob. over $V$ | LM-fused teacher distribution (support restricted to top-$K_{\text{tok}}$). |
| $\mathbb{K}_t$ | set $\subseteq \{1, \ldots, V\}$ | Top-$K_{\text{tok}}$ token ids from $\boldsymbol{Z}_t$ (after masking specials). |
| ctx | list (length $M$) | Document context: last $M$ accepted lines (no ground truth). |
| $K_{\text{tok}}$ | scalar | Token shortlist size for LM fusion (per step). |
| $B$ | scalar | *Beam size* for decoding/reranking (size of the $n$-best list). |
| $\beta$ | scalar | LM weight in sequence-level shallow fusion (reranking). |
| $\alpha$ | scalar | LM weight in token-level fusion (distillation teacher). |
| $\tau$ | scalar | Temperature for distillation (and logits in fusion). |
| $\delta$ | scalar | Entropy gate threshold for updates (uncertainty-based). |
| $\hat{\boldsymbol{S}} \in [0,1]^{H' \times W'}$ | map | Predicted soft skeleton (stroke head output). |
| $\boldsymbol{S}^{\star} \in [0,1]^{H' \times W'}$ | map | Target skeleton from $g(\boldsymbol{X})$ (invert $\rightarrow$ single erosion $\rightarrow$ resize to $(H', W')$). |
| $g(\cdot)$ | transform | Deterministic image-to-skeleton pipeline used to form $\boldsymbol{S}^{\star}$. |
| $\tau_{\text{skel}}$ | scalar | Binarization threshold for skeleton maps in the Chamfer term. |
| $\gamma$ | scalar | Weight on Chamfer distance in $\mathcal{L}_{\text{skel}}$. |
| $\epsilon$ | small $> 0$ | Smoothing constant in Dice. |
| $\mathcal{L}_{\text{skel}}$ | loss | Dice + $\gamma \cdot$Chamfer for skeleton alignment. |
| $\mathcal{L}_{\text{LM}}$ | loss | Uncertainty-gated distillation to the LM-fused teacher. |
| $\mathcal{L}_{\text{TTA}}$ | loss | $\lambda_{\text{skel}} \mathcal{L}_{\text{skel}} + \lambda_{\text{LM}} \mathcal{L}_{\text{LM}}$. |
| $\lambda_{\text{skel}}, \lambda_{\text{LM}}$ | scalars | Loss weights for skeleton / LM terms. |
| $\mathcal{R}(\hat{\boldsymbol{y}}^{(b)})$ | score | Sequence-level reranking score for beam $b$: model score + $\beta \cdot$(length-normalized LM score). |
| $\mathbb{B} = \{\boldsymbol{X}_i\}_{i=1}^N$ | set | Test mini-batch (size $N$). |
| $N$ | scalar | Batch size ($N = |\mathbb{B}|$). |
| $\eta$ | scalar | Inner-loop learning rate. |

