# OpenReview forum: "SCRIBE: STROKE- AND CONTEXT-REGULARIZED TEST-TIME ADAPTATION FOR HANDWRITTEN TEXT RECOGNITION"
_ICLR.cc/2026/Conference — Submitted to ICLR 2026_

### Official Review · Reviewer_p3MG · 2025-10-27

**Soundness:** 1
**Presentation:** 3
**Contribution:** 2
**Rating:** 2
**Confidence:** 4

**Summary:**

This paper proposes SCRIBE, a framework for test-time adaptation of handwritten text recognition OCR models, using unpaired data from the target domain to improve the performance. The experiments are done on the TrOCR family of models, using up to 5 updates on a batch of 8 samples of unlabeled test data. The adaptation has 3 main components:

1. A "skeleton decoder" head attached to the visual encoder, enforcing consistency of the features coming out of the encoder with the skeleton extracted from the original image (the metric used is Chamfer pairwise set distance + IoU-type Dice distance on the set of pixels in the boolean masks). The skeletonization is done using erosion + dilation..

2. LM-Guided KL regularization with GPT-2 LM, where the probability distribution coming from the model is fused with probabilities from the LM - KL divergence term between this distribution and the one from the model decoder pushes the decoder to produce distribution closer to the fused one.

3. LM Rescoring (post-adaption) - During decoding, beam search elements are re-ranked based on fusing the score with the score coming from the language model (GPT-2).

The evaluation is done on 4 datasets (IAM, RIMES (French), GW (18th century English letters), Bentham (19th century English manuscripts)). The comparison is done to a number of other HTR models (much worse than the base TrOCR model). The ablation study compares to the base TrOCR model itself as well as leave-one-out on 3 suggested components. Comparison to other, simpler TTA approaches is performed. Comparison of fine-tuning and fine-tuning with TTA is performed in appendix C.

**Strengths:**

Originality: The individual ideas (LM rescoring, skeleton losses, TTA) are not new. The originality lies only in the specific combination of a geometric loss and an LM-based loss within a TTA gradient loop for HTR.

Quality & Clarity: The paper is clearly written and easy to understand, the proposed approaches show some improvements compared to the baseline model, the ablation study studies the components well. However, the actual results presented by the authors (Table 3) undermine the paper's main narrative about the need for joint visual and textual test-time adaptation, furthermore there is significant risk of contamination affecting the results (hence the soundness score. See 'Weaknesses' below).

Significance: Between the results of the paper suggesting that the central idea of joint visual+text TTA is not necessarily the main driver of the results, the risks of possible contamination, and absence of details on the computational complexity of the proposed approach, the significance is limited.

**Weaknesses:**

The main weakness of the paper is the disconnect between the central claim about the usefulness of visual-textual TTA and the presented results:

1. The results in Table 3 suggest that effect of visual adaptation is minimal (removing it minimally increases the WER) and that generally the effect of using LLM for re-scoring the beam-search decoding (which is not TTA) is the strongest among all 3 components. The ablation study could be significantly strengthened by showing the results of the baseline that does not use any TTA but only uses LLM-based rescoring for decoding.

2. The datasets used for evaluation, in particular IAM (which is 25+ years old)  and RIMES (which is publicly available on HuggingFace / Kaggle since a long time) are quite likely present in the pre-training corpus of the GPT-2 model, which makes it impossible to decouple the effect of using the LM in general, from the result of training data contamination. The evaluation could be significantly strengthened by evaluation on the datasets released recently and with larger dataset drift than the English-French (ex. math handwriting recognition based on handwritten portion of UNIMER-1M or MathWriting).

3. The computational burden of the proposed approach is not discussed, but given that TrOCR-base is ~224M and TROCR-large is~493M parameters, while GPT-2 Large is ~774M parameters, this likely increases the computational footprint by a factors of 2.5x-5x. Currently results from Table 3 shows that improvement from going TrOCR-base to TrOCR-large (5.1->3.6 CER) helps more than adding the proposed approach (5.1->3.7 CER), while also being more parameter-efficient.

To sum up, in my opinion, for this paper to be a significant contribution, it needs to show:
- That TTA is actually brining measurable improvement on top of the LLM-based reranking with beam-decoding
- Performance on datasets that have clearly not contaminated the LLM pre-training corpus
- That increase in computational footprint is justified.

**Questions:**

No additional questions, but I would love for the authors to address the 3 points I've outlined in the "Weaknesses" section.

---

> ### Author Response · Authors · 2025-11-21
> **Response to Reviewer p3MG(1/3)**
>
> **We appreciate your valuable time and feedback to make our paper stronger. We have revised the paper to incorporate all of your suggested changes, and all modifications are highlighted in blue in the updated version for your easy inspection.**
>
> >***Weakness1:***
>
> **On the “disconnect” between the visual–textual TTA claim and the results / need for LM-only baseline**
>
> Thank you for this insightful comment. We agree that Table 3 alone could give the impression that LM reranking is the dominant component, and that a “LM-only, no TTA” baseline is important. We have therefore added exactly this baseline and re-examined the role of visual adaptation.
>
> 1. New baseline: frozen TrOCR + LM-only (no TTA)
> We now evaluate a setting where we do not adapt the HTR model at all and only apply LLM-based rescoring at decoding time (beam search + GPT-2 scoring, no skeleton, no LM-KL, no gradient updates). the revised version have add this result in Table 3.  We obtain:
>
>
> |                      |      IAM      |     RIMES     |      GW       |    Bentham    |
> |:--------------------:|:-------------:|:-------------:|:-------------:|:-------------:|
> |  Frozen TrOCR   | 11.78 / 35.86 | 27.54/ 65.25 | 16.86/ 29.07 | 17.55/ 28.02 |
> | TrOCR + LM-only | 10.08/ 21.06 | 21.80/ 60.84 | 13.51/ 26.83 | 13.49/24.50 |
> |   Our full pipline   |  8.06/17.65  | 21.29/ 59.04 | 12.40/ 25.40 | 12.31/ 22.59 |
>
>
>
> This shows:
>
> on four datasets, LM-only does help over the frozen baseline, confirming your intuition that LLM-based rescoring is a strong component.
>
> **However, full TTA still improves consistently over LM-only:**
>
>
> In other words, LM-only decoding is a strong and necessary baseline—but it does not reach our final performance. LM reranking is indeed a strong linguistic component, but visual adaptation (skeleton) and LM-KL inside the gradient loop still provide additional error reduction beyond LM-only on all four datasets.
>
> We will add the “TrOCR + LM-only” rows in the revised version and clarify that our central claim is not that LM rescoring alone is new, but that combining a topology-aware skeleton loss (imported from medical imaging) with LM-based guidance inside a source-free TTA loop for HTR yields the best CER/WER and goes significantly beyond pure LM rescoring.

---

> > ### Author Response · Authors · 2025-11-21
> > **Response to Reviewer p3MG(2/3)**
> >
> > >***Weakness2:***
> > **On possible GPT-2 contamination of IAM/RIMES and the need for newer, drifted datasets**
> >
> > We agree that IAM (25+ years old) and RIMES are likely to appear in generic web corpora and that contamination is a valid concern for GPT-2. To address this, we took several steps and added new experiments showing that our gains are not just a consequence of GPT-2 memorizing these benchmarks.
> >
> > **New “drifted” datasets as Reviewer p3MG suggested: MathWriting and UniMER-1M (HWE):**
> > Following the reviewer’s suggestion, we evaluated on recent math handwriting datasets that are very unlikely to be present in GPT-2’s training corpus and exhibit a much stronger domain shift than English–French report in (CER/WER):
> >
> >
> >
> > |                                    |    MathWriting     |     UniMER-1M      |
> > |:----------------------------------:|:------------------:|:------------------:|
> > |        TrOCR-base(Baseline)        | 88.05%/  334.95% | 85.89%/  98.38% |
> > | Our Method | 85.19%/  207.89% | 82.17%/ 97.18% |
> >
> >
> >
> > Both datasets are recent and math-heavy, yet our visual+text TTA still yields **consistent CER/WER gains**, which is hard to explain by contamination.
> >
> > **LMs that do not include the target data: CroissantLLM and OLMo2:**
> > We also replace GPT-2 with language models whose training corpora do not include our evaluation sets, report in (CER/WER):
> >
> >
> >
> > |                                  |      IAM       |     RIMES      |
> > |:--------------------------------:|:--------------:|:--------------:|
> > |      TrOCR-stage1 baseline       | 11.78%/ 35.86% | 23.32%/ 55.30% |
> > | Stage1 + CroissantLLM(French LM) |      -  -      | 17.57%/ 47.41% |
> > |    stage1 + OLMo2(English LM)    | 8.08%/ 17.78%  |     -   -      |
> > |    Our Method    | 8.06%/ 17.65%  |     17.93%/ 48.39%      |
> >
> >
> > **RIMES + CroissantLLM (French LM without RIMES)**
> >
> >
> >
> > Stage1 + CroissantLLM (LM reranking only, no TTA): CER 17.57, WER 47.41
> > → A strong improvement, even though the LM has never seen RIMES.
> >
> > **IAM + OLMo2 (new English LM without IAM)**
> >
> > The fact that OLMo2 (which does not include IAM) produces essentially the same CER/WER as GPT-2-Large under our TTA framework indicates that the gains are not tied to GPT-2’s pretraining corpus, but come from using a reasonably strong LM prior inside TTA.
> >
> > **Mitigations on the vision side and scope of our claim:**
> >
> > On IAM, we deliberately use TrOCR-stage1 (pretraining only, no fine-tuning on IAM) to **avoid visual-side contamination** from public TrOCR checkpoints.
> >
> > Our method also yields consistent gains on GW and Bentham, historical datasets that are extremely unlikely to be in GPT-2’s training data.
> >
> > In the revised version we have **soften any over-strong wording** and explicitly acknowledge that some text-side contamination on IAM/RIMES is possible, but the pattern of **improvements across** **MathWriting, UniMER-HWE, GW, Bentham, CroissantLLM on RIMES, and OLMo2 on IAM** shows that our visual+text TTA remains effective even with strong **datasets domain shift** and **recent LMs that are not contaminated by these benchmarks**.
> >
> > **Overall, these additional experiments support that our gains are due to general LM priors combined with visual TTA, rather than benchmark memorization by GPT-2.**

---

> > > ### Author Response · Authors · 2025-11-21
> > > **Response to Reviewer p3MG(3/3)**
> > >
> > > >***Weakness3:***
> > >
> > > **On computational burden and comparison to scaling TrOCR-base**
> > >
> > > Thank you for raising the question of computational footprint. We agree that parameter count and latency are important in HTR settings. Our goal is not to claim that the heaviest variant is always practical, but to provide a **family of operating points** with different accuracy–efficiency trade-offs.
> > >
> > > **Latency: full pipeline vs. lighter variants**
> > >
> > >
> > >
> > > |        Column 1         |  Skeleton TTA  |  LM-KL   | LM reranking/Decoding |
> > > |:-----------------------:|:--------------:|:---------------:|:---------------------:|
> > > | Baseline TrOCR decoding |       - -       |       - -        |    103.54 ms/line     |
> > > |  Ours (LN/bias-only)   | +7.98 ms/line  | +42.27 ms/line  |    +12.68 ms/line    |
> > > |     Ours (Full update)      | +11.00 ms/line | +288.49 ms/line |    +14.23 ms/line     |
> > >
> > >
> > >
> > >
> > > Above table summarizes the latency breakdown on IAM (single GPU, averaged over 100 lines). Baseline TrOCR decoding runs at about **103.54 ms** per line. Our full TTA pipeline (skeleton + LM-KL + LM reranking with GPT-2-Large and all parameters updated) reaches roughly **417 ms per line**, about 4× the baseline, with most of the overhead coming from the **LM-KL term**. The **LN/bias-only variant** keeps the same TTA logic but **updates fewer than 0.2%** of the parameters and **runs at around 166 ms per line**, about **1.6× the baseline**, which is much **closer to a realistic deployment setting**. If needed, we can also use a smaller LM (e.g., DistilGPT-2), which further reduces the cost (**≈1.8× baseline**) at the price of a small accuracy drop.
> > >
> > > **Model parameters: how much do we really “add”?**
> > >
> > > Our HTR backbone remains **TrOCR-base (≈334M params)**, and during TTA we **only update LayerNorm and bias terms**, i.e.,** <0.2% of the weights (≈4×10⁵ / 3.34×10⁸)** without changing the architecture. The external LM (e.g., GPT-2-Large) is a separate shared module that can run server-side, so from the HTR model’s perspective our adaptation is **extremely parameter-efficient**.
> > >
> > > **Accuracy–efficiency trade-off is favorable** report in(CER/WER)
> > >
> > >
> > >
> > > |     TTA Methods     |    IAM     |    RIMES    |     GW      |   Bentham   |
> > > |:-------------------:|:----------:|:-----------:|:-----------:|:-----------:|
> > > |     TrOCR-base      | 5.10/ 11.95 | 27.54/ 65.25 | 16.86/ 29.07 | 17.55/ 28.02 |
> > > | Ours (LN/bias-only) | 3.68/ 10.94 | 21.46/ 59.53 | 12.91/ 25.84 | 12.94/ 23.30 |
> > > |     Ours (Full update)      | 3.61/ 10.88 | 21.29/ 59.04 | 12.40/ 25.40 | 12.31/ 22.59 |
> > >
> > >
> > >
> > >
> > >
> > > On TrOCR-base, our heaviest GPT-2-Large TTA improves the frozen baseline. The LN/bias-only variant (≈1.6× runtime) recovers **≈88–97%** of the full GPT-2-Large gains while updating <0.2% of parameters and adding only **a modest 1.6× latency** overhead.
> > >
> > > **In summary,** while the maximal configuration is indeed compute-heavy and best suited for offline or batch scenarios, we explicitly provide and evaluate lighter variants **(LN/bias-only, or smaller LM, or fewer TTA steps)** that match the parameter budget of TENT/EATA, keep latency in the **≈1.6× range**, and still deliver **strong CER/WER improvements**. We have highlight this speed–accuracy trade-off more clearly **in the revised paper Table 2** to emphasize that our method is designed to be tunable and practically deployable, not tied to a single heavy setting.  **In the revised version we have add these experiment in the table 2.**
> > >
> > >
> > >
> > > >***Weakness4:***
> > > **Clarifying the numbers: TrOCR-base vs. TrOCR-large vs. our TTA**
> > >
> > > The reviewer p3MG comment “Table 3 shows that improvement from going TrOCR-base to TrOCR-large (5.1 → 3.6 CER) helps more than adding the proposed approach (5.1 → 3.7 CER)” is based on a misunderstanding of which numbers correspond to which models.
> > >
> > > **From Table 1 (IAM, TrOCR backbones) in paper**, the correct values are:
> > >
> > >
> > >
> > >
> > > |                       |         IAM         |
> > > |:---------------------:|:-------------------:|
> > > |  TrOCR-base (frozen   | CER 5.10, WER 11.95 |
> > > | TrOCR-base + our TTA  | CER 3.61, WER 10.88 |
> > > | TrOCR-large (frozen)  | CER 3.71, WER 9.58  |
> > > | TrOCR-large + our TTA | CER 3.17, WER 8.78  |
> > >
> > >
> > > Going from **TrOCR-base → TrOCR-large** (both frozen) gives
> > > **5.10 → 3.71 CER** (ΔCER = **−1.39**).
> > >
> > >
> > > Applying our TTA on TrOCR-base gives
> > > **5.10 → 3.61** CER (ΔCER = **−1.49**).
> > >
> > >
> > > In other words:
> > >
> > >
> > > The **3.6 CER** number cited by the reviewer is **actually TrOCR-base + our method**, not TrOCR-large.
> > >
> > >
> > > **TrOCR-base + our TTA (3.61)** is already better than **TrOCR-large without TTA (3.71)**, even though the backbone is smaller.
> > >
> > >
> > > When we also apply TTA to TrOCR-large, **we obtain the best result (3.17 CER)**. Thus, the table shows the opposite of what is claimed in the review.

---

> ### Comment · Reviewer_p3MG · 2025-11-24
>
> I thank the authors for this set of additional experimental results - it is very impressive that you have been able to address all of those to some degree.
>
> For the first set of results about the TTA effects: They are very useful, but in my opinion they point in the same direction - the LM-only component closes 80+% of the gap to the full pipeline on RIMES, GW, and Bentham datasets, thus confirming that the TTA is a minor addition.
>
> For the second set of results regarding the datasets with domain drift, and model memorization: Thank you for the additional results with the newer models. This is a strong signal. The numerical results on the math datasets are not representative though, because CER of 88% means every 8 out of 9 characters are wrong, and improvement from 85% to 82% can not be meaningfully attributed to the better model.
>
> For the third set of results regarding latency, thank you for pointing out the confusion on my side, where 3.61 is actually the number with TTA on base, and 3.71 is the number with frozen on large. I do believe that the point still stands about the ability to achieve similar numbers without algorithmic improvements and while maintaining similar latency. I thank authors for highlighting the results with 1.6 latency factor vs 4x and I believe this strengthens the paper.
>
> Overall, these additional results from the authors are useful, helpful, and increase the quality of the work, and I would be happy to increase my rating to 3 or 4 - but based on the actual insights from these numbers, I believe significance of the proposed approach is still limited, thus targeting 'reject'-side ratings (the final rating will depend on the discussions with other reviewers)

---

> > ### Author Response · Authors · 2025-11-26
> > **Reply to Reviewer p3MG**
> >
> > Dear Reviewer **p3MG**,
> >
> > Thank you for your prompt reply and for providing additional comments. We are glad to hear that some of your concerns have been resolved and that you are considering increasing your rating.
> >
> > Regarding your remaining concerns, we would like to offer additional clarifications:
> >
> > ### **1. Clarification on the TTA-effect results**
> >
> > We would like to clarify that the observation that the **LM-only component closes 80%+ of the gap does *not*** imply that the TTA module contributes only the remaining 20%. Performance gains from different components are **not linearly additive**. For example, even if Method A and Method B each independently yield a 50% improvement over the SOTA, their combination does not necessarily produce a 100% improvement.
> >
> > To better illustrate this point, we have reorganized the ablation study into the table shown below. As demonstrated, **both components contribute comparably to the final performance improvement**, especially in terms of CER on IAM/RIMES/GW.
> >
> >
> > |                        | IAM          |    RIMES     |      GW      |   Bentham    |
> > |:----------------------:| ------------ |:------------:|:------------:|:------------:|
> > |      Frozen TrOCR      | 11.78/ 35.86 | 27.54/ 65.25 | 16.86/ 29.07 | 17.55/ 28.02 |
> > |    TrOCR + LM-only     | 10.08/ 21.06 | 21.80/ 60.84 | 13.51/ 26.83 | 13.49/24.50  |
> > | TrOCR + TTA-only  | 10.44/24.08 | 21.87/61.61 | 14.06/28.35 | 14.19/26.42 |
> > |    Our full pipline    | 8.06/ 17.65   | 21.29/ 59.04 | 12.40/ 25.40 | 12.31/ 22.59 |
> >
> >
> > ### **2. Regarding the second set of results on datasets with domain drift and model memorization**
> >
> > We believe there may be a misinterpretation here. We fully agree that the absolute CER on MathWriting and UniMER-1M is high. This is expected because these datasets require **image → LaTeX** transcription, which differs fundamentally from TrOCR’s training data. TrOCR is pretrained on **natural handwriting OCR (plain text)**, whereas the math datasets introduce both a substantial **input-domain shift** (mathematical notation, layout structures) and a substantial **output-space shift** (LaTeX token sequences rather than natural language). Thus, the high absolute errors reflect **task mismatch**, not model memorization.
> >
> > Our intention in including these datasets was to directly address **contamination concerns raised by you**. These datasets are recent, math-specific, and extremely unlikely to appear in GPT-2’s pretraining corpus. Under such conditions, where neither TrOCR nor GPT-2 has seen similar input or output distributions, our method still produces **consistent relative CER/WER improvements**, suggesting that the performance gains come from the **TTA mechanism**, not from dataset memorization.
> >
> >
> > ### **3. Regarding the third set of results on latency**
> >
> > We are glad that our response is recognized as adding further strength to the paper. We would like to emphasize that our primary goal is to explore source-free test-time adaptation to improve performance, rather than to optimize efficiency. Developing a more refined trade-off between performance and latency is an important direction, and we consider it valuable future work.
> >
> >
> > ---
> >
> > We sincerely thank you again for your time and valuable feedback. We greatly appreciate that you are considering increasing the rating. We hope this additional clarification helps address the remaining concerns. If anything is still unclear, we are happy to continue the discussion.

---

### Official Review · Reviewer_YtZG · 2025-10-30

**Soundness:** 2
**Presentation:** 3
**Contribution:** 2
**Rating:** 4
**Confidence:** 4

**Summary:**

This paper focus on test-time adaptation for HTR. The authors present a label-free and source data-free test-time adaptation technique that enhances the generalization of HTR models on unseen domains by jointly optimizing a lightweight stroke-structure loss and a document-conditioned language prior. Experiments demonstrate the effectiveness of the proposed method.

**Strengths:**

1.	This approach involves updating models online on unlabeled target data. It does so without source replay or target-only fine-tuning, which is an interesting idea.

2.	The proposed method achieves competitive results.

**Weaknesses:**

1.	The author fails to clearly elaborate on whether the Test-time adaptation of HTR is still worthy of research in the era of Multimodal Large Language Models (MLLMs), and whether MLLMs have achieved sufficient generalization across different scenarios such that further research on the test-time adaptation of HTR is unnecessary.

2.	The foundation model (eg GPT-2) being used is relatively old. The foundation model used, TrOCR, is two years old, and the language model, GPT-2, is also relatively outdated.

3.	HTR models typically have strict requirements regarding model parameter and inference speed. However, the LM reranking method proposed in this paper substantially increases the model parameter and reduces inference speed, which may limit its practical applicability.


4.	The authors describe the method as plug-and-play, yet it has only been validated on TrOCR and lacks validation on many other relative models.

**Questions:**

Please see the Weakness section.

---

> ### Author Response · Authors · 2025-11-21
> **Response to Reviewer YtZG(1/3)**
>
> **We appreciate your valuable time and feedback to make our paper stronger. We have revised the paper to incorporate all of your suggested changes, and all modifications are highlighted in blue in the updated version for your easy inspection.**
>
> >***Weakness1:***
>
> **On the relevance of HTR Test-Time Adaptation in the era of MLLMs**
>
> Thank you for raising this broader question. We agree that Multimodal LLMs (MLLMs) are reshaping the landscape, and we have started to evaluate them as baselines. However, our experiments and the model sizes both indicate that specialized HTR + source-free TTA remains highly relevant and complementary, not obsolete. In the revised version, we add a **new Table 5 that compares recent and popular generic OCR/VLM baselines on IAM**.
>
> Empirical comparison on IAM, we evaluated a newest 4B-parameter MLLM and a much smaller HTR model:
>
>
>
>
> |                  | TrOCR-base(Our Method) |  Qwen3-VL-4B  |        Donut         |
> | ---------------- |:----------------------:|:-------------:|:--------------------:|
> | baseline         |     5.10%/ 11.95%      | 7.68%/ 31.73% |    28.0%/  44.0%     |
> | After Adaptation |     3.61%/  10.88%     | 7.38%/ 30.96% | 27.09%/     42.35% |
>
> **Qwen3-VL-4B-Instruct (≈4B parameters, multimodal): CER 7.68%, WER 31.73%**
>
> **TrOCR-base + our TTA (≈334M parameters, ≈12× smaller): CER 3.61%, WER 10.88% (Table 1)**
>
> (For reference, document-oriented models like Donut are much worse: CER 28.0%, WER 44.0% on IAM.)
>
> Thus, **a 334M HTR model with our source-free TTA clearly outperforms a 4B-parameter MLLM on the same dataset**, especially in WER (31.7% → 10.9%). **MLLMs have not yet “solved”** line-level HTR at the accuracy level required by our setting.
>
> **Model design, cost, and deployment.**
> Qwen3-VL-4B is a general-purpose multimodal reasoner, not a pixel-precise line recognizer. It is much heavier in parameters, cost, and latency, and difficult to deploy in scanners, archives, or on-device HTR. In contrast, our TrOCR-base + TTA pipeline uses a compact 334M backbone, adapts only at test time with no source data or target labels, and can run in an LN/bias-only mode that updates <0.2% of parameters with only ≈1.6× baseline runtime, making source-free HTR TTA practically deployable.
>
> **Complementarity, not redundancy.**
> In realistic systems, MLLMs typically consume text, not raw pixels, so a strong HTR front-end remains necessary. Our stroke- and LM-regularized TTA improves transcript quality for downstream QA/retrieval rather than being replaced by MLLMs.
>
> **In the revision**, Table 5 now include explicit comparisons between MLLM and TrOCR-based baselines, showing that a 334M TrOCR-base model with our TTA substantially outperforms a 4B-parameter MLLM on IAM while being cheaper and easier to deploy. This indicates that MLLMs have not made HTR TTA obsolete; our method is a practical and complementary way to improve recognition across domains and writers.

---

> > ### Author Response · Authors · 2025-11-21
> > **Response to Reviewer YtZG(2/3)**
> >
> > >***Weakness2:***
> > >
> > **On using “old” foundation models (TrOCR, GPT-2)**
> > Thank you for this comment. Our goal is not to propose a new foundation model, but a source-free TTA framework that can sit on top of a HTR backbone and LM. We initially chose TrOCR and GPT-2 because they are strong, widely-used baselines with stable open-source implementations. To check whether our conclusions still hold with newer models, we have run additional experiments on IAM:
> >
> > **New 2025 LM (OLMo2) with our TTA:**
> >
> >
> >
> >
> >
> > |              NEW LM              |     IAM      |
> > |:--------------------------------:|:------------:|
> > |    TrOCR-stage1 + TTA + OLMo2    | 8.17 / 17.78 |
> > | TrOCR-stage1 + TTA + GPT-2-Large | 8.06 / 17.65 |
> > |     TrOCR-base + TTA + OLMo2     | 3.67 / 10.72 |
> > | TrOCR-stage1 + TTA + GPT-2-Large | 3.61 / 10.88 |
> >
> >
> >
> > TrOCR-stage1 + TTA + OLMo2: CER 8.17%, WER 17.78% (vs. 8.06 / 17.65 with GPT-2-Large – essentially identical)
> >
> > TrOCR-base + TTA + OLMo2: CER 3.67%, WER 10.72% (vs. 3.61 / 10.88 with GPT-2-Large – very similar overall)
> >
> > **New 2025 foundation MLLM baseline:**
> >
> >
> >
> > |   NEW Baseline    |        IAM        |
> > |:-----------------:|:-----------------:|
> > |    Qwen3-VL-4B    |    7.68/ 31.73     |
> > | Qwen3-VL-4B + LM  | 7.39/ 30.97 |
> > |    TrOCR-base     |    5.10/ 11.95     |
> > | TrOCR-base + LM | 3.61/ 10.88  |
> >
> >
> > Qwen3-VL-4B + LM reranking on IAM: CER 7.39%, WER 30.97%
> >
> > Our TrOCR-base + TTA (≈334M params, ~12× smaller than Qwen3-4B): CER 3.61%, WER 10.88%
> >
> > So even when we plug in newer LMs (OLMo2) or a newer, much larger MLLM (Qwen3-VL-4B), they do not outperform our proposed HTR+TTA pipeline; if anything, they reinforce the main message: our gains are not tied to a particular “old” LM (GPT-2 vs. OLMo2 gives almost the same CER/WER), and a specialized 334M HTR backbone with our TTA still clearly outperforms a 4B-parameter modern MLLM on IAM.
> >
> > In the revised version **we add these experiment in table 5 and in Appendix table 10 and table 11** and clarify that our contribution is a model-agnostic TTA framework, and we report these new OLMo2 and Qwen3-VL-4B results to show that the method remains effective and competitive even when newer foundation models and LMs are considered.

---

> > > ### Author Response · Authors · 2025-11-21
> > > **Response to Reviewer YtZG(3/3)**
> > >
> > > >***Weakness3:*** **On parameter count and inference speed of LM reranking**
> > >
> > > We agree that many HTR applications have strict constraints on model size and latency. Our goal is not to claim that the heaviest variant is always practical, but to provide a family of operating points with different accuracy–efficiency trade-offs.
> > >
> > > **Latency: full pipeline vs. lighter variants**
> > >
> > >
> > >
> > > |        Column 1         |  Skeleton TTA  |  LM-KL   | LM reranking/Decoding |
> > > |:-----------------------:|:--------------:|:---------------:|:---------------------:|
> > > | Baseline TrOCR decoding |       - -       |       - -        |    103.54 ms/line     |
> > > |  Ours (LN/bias-only)   | +7.98 ms/line  | +42.27 ms/line  |    +12.68 ms/line    |
> > > |      Ours (Full update)      | +11.00 ms/line | +288.49 ms/line |    +14.23 ms/line     |
> > >
> > >
> > >
> > >
> > > Above table summarizes the latency breakdown on IAM (single GPU, averaged over 100 lines). Baseline TrOCR decoding runs at about **103.54 ms** per line. Our full TTA pipeline (skeleton + LM-KL + LM reranking with GPT-2-Large and all parameters updated) reaches roughly **417 ms per line**, about 4× the baseline, with most of the overhead coming from the **LM-KL term**. The **LN/bias-only variant** keeps the same TTA logic but **updates fewer than 0.2%** of the parameters and **runs at around 166 ms per line**, about **1.6× the baseline**, which is much **closer to a realistic deployment setting**. If needed, we can also use a smaller LM (e.g., DistilGPT-2), which further reduces the cost (**≈1.8× baseline**) at the price of a small accuracy drop.
> > >
> > > **Model parameters: how much do we really “add”?**
> > >
> > > Our HTR backbone remains **TrOCR-base (≈334M params)**, and during TTA we **only update LayerNorm and bias terms**, i.e.,** <0.2% of the weights (≈4×10⁵ / 3.34×10⁸)** without changing the architecture. The external LM (e.g., GPT-2-Large) is a separate shared module that can run server-side, so from the HTR model’s perspective our adaptation is **extremely parameter-efficient**.
> > >
> > > **Accuracy–efficiency trade-off is favorable**
> > >
> > >
> > >
> > > |     TTA Methods     |    IAM     |    RIMES    |     GW      |   Bentham   |
> > > |:-------------------:|:----------:|:-----------:|:-----------:|:-----------:|
> > > |     TrOCR-base      | 5.10/ 11.95 | 27.54/ 65.25 | 16.86/ 29.07 | 17.55/ 28.02 |
> > > | Ours (LN/bias-only) | 3.68/ 10.94 | 21.46/ 59.53 | 12.91/ 25.84 | 12.94/ 23.30 |
> > > |     Ours (Full update)      | 3.61/ 10.88 | 21.29/ 59.04 | 12.40/ 25.40 | 12.31/ 22.59 |
> > >
> > >
> > >
> > >
> > >
> > > On TrOCR-base, our heaviest GPT-2-Large TTA improves the frozen baseline. The LN/bias-only variant (≈1.6× runtime) recovers **≈88–97%** of the full GPT-2-Large gains while updating <0.2% of parameters and adding only **a modest 1.6× latency** overhead.
> > >
> > > **In summary,** while the maximal configuration is indeed compute-heavy and best suited for offline or batch scenarios, we explicitly provide and evaluate lighter variants **(LN/bias-only, or smaller LM, or fewer TTA steps)** that match the parameter budget of TENT/EATA, keep latency in the **≈1.6× range**, and still deliver **strong CER/WER improvements**. We have highlight this speed–accuracy trade-off more clearly **in the revised paper Table 2** to emphasize that our method is designed to be tunable and practically deployable, not tied to a single heavy setting.  **In the revised version we have add these experiment in the table 2.**
> > >
> > >
> > > >***Weakness4:***
> > > **On the “plug-and-play” claim and validation beyond TrOCR**
> > >
> > > Thank you for pointing this out. We agree that, as currently phrased, calling our full method “plug-and-play” is too strong given that our full TTA pipeline (skeleton + LM-KL + LM reranking) has only been validated on TrOCR-style encoder–decoder HTR backbones.
> > >
> > > Our intention was more modest:
> > >
> > > At test time, our framework does not require any additional source data or re-training from scratch, and some components (in particular LM reranking) can indeed be attached in a plug-and-play fashion to different sequence models.
> > >
> > > To address this concern, in the revised version we will:
> > >
> > > **Tone down the wording**: we will remove the generic “plug-and-play” claim and instead describe our method as a source-free test-time adaptation framework instantiated on TrOCR-style HTR models.
> > >
> > > Importantly, **our main contributions and empirical gains do not rely on the “plug-and-play” label.** We have updated the text accordingly so that the scope of our claims is fully aligned with the experiments currently reported.

---

### Official Review · Reviewer_QrPe · 2025-10-31

**Soundness:** 3
**Presentation:** 3
**Contribution:** 3
**Rating:** 6
**Confidence:** 4

**Summary:**

The authors proposed an interesting approach of test-time adaptation for handwritten text recognition. This application normally suffers from distribution shift, for example, new writers, new historical substrates, a lot of noise, different layouts, and languages. The idea is to couple a stroke-preserving geometric with a document-conditioned language. The paper shows that the approach improves TrOCR performance across four benchmarks.

The proposed label-free skeleton loss uses an uncertainty-gated KL for a document-aware LM teacher, they also added a sequence-level LM reranking, which offers an interesting combination for source-free that adapts optics recognition without language or applies LM guidance without geometric anchoring. I see this as the main contribution.

The idea may help with operational constraints of handwritten text recognition and privacy-sensitive uses.

**Strengths:**

I liked the idea of a stroke-preserving self-supervision on an auxiliary skeleton head, especially the Dice and Chamfer loss. The document-conditioned language proposed is using both loss as an uncertainty-gated KL distillation term, which helps a kind of sequence-level shallow-fusion reranking of the outputs and improving performance. The authors should explore this part in the paper.

The paper also demonstrates consistent gains on IAM, RIMES, GW, and Bentham. The interesting part is that they have the same budget and without a source reply.

The main idea is to couple visual topology preservation with linguistic regularization and this occurs at both token and sequence levels.

Also liked the losses used, the Dice+Chamfer and uncertainty-gated LM-KL. The strategy for sensible gating and the reset strategy help to limit drift.

The authors perform ablations to show the component’s effect.

The experimental results show gains on TrOCR backbones over multiple datasets.

**Weaknesses:**

Not sure I understand correctly, but in Table 2, the proposed method adapts all encoder+decoder parameters, while TENT/EATA baselines adapt LN/bias only. If it is true, that is an unfair comparison that can inflate margins.

Table 3 removes components from the full system but never shows frozen TrOCR + LM reranking or frozen TrOCR + LM-KL, which seems to have no weight updates. Without this it is not easy to understand which component is responsible for the results, part of the improvement could be from better decoding rather than adaptation. Please add frozen + LM-rerank and frozen + LM-KL rows and also quantify the net “adaptation-only” delta.

I don't understand the description of Fig. 3 that alternately describes erosion, centerline/contour overlays, and cite thinning in that context, these are not equivalent and affect the claimed topology preservation.

I believe there is some backbone inconsistency across datasets, as IAM is evaluated with stage-1 while others use base (and sometimes small/large), which may confound average gains.

Please add confidence intervals/repeated runs to evaluate the statistical significance of the results.

Improve the methodology description, as some reproducibility hyperparameters (λ’s, α, β, τ, K, δ, skeleton resolution/threshold) are missing from the text and are needed for reproducibility.

The claim on novelty also needs a better discussion, the “first source-free HTR TTA” could be regarded as overreaching since Tula’23 and Gu’25 also present contributions in this direction. This could be better addressed by the authors in the discussion.

**Questions:**

The experiments only addressed Latin-script datasets. Do you think performance will be maintained with a non-Latin corpus?

What is the delta from frozen TrOCR + LM reranking and frozen + LM-KL (no gradient) relative to full TTA?

Why does GPT-2 large outperform CamemBERT on RIMES? Please compare with a French causal LM of similar capacity and detail tokenization/normalization.

---

> ### Author Response · Authors · 2025-11-21
> **Response to Reviewer QrPe(1/5)**
>
> **We appreciate your valuable time and feedback to make our paper stronger.  We have revised the paper to incorporate all of your suggested changes, and all modifications are highlighted in blue in the updated version for your easy inspection.**
>
> >***Weakness1:***
>
> **Reviewer concern (fairness of adaptation).**
> **The reviewer states that in Table~2, our method ``adapts all encoder+decoder parameters, while TENT/EATA baselines adapt LN/bias only,'' which would indeed be unfair if true.**
>
> We thank the reviewer for pointing this out. You are correct that, in Table 2, our default variant adapts all encoder+decoder parameters, whereas TENT and EATA only update LayerNorm and bias parameters. To address this, we have added a new experiment where our method is **restricted to exactly the same parameter** subset as TENT/EATA (LN + bias only). Also update this result in Table 2 in the revised version.
>
> Concretely, for a TrOCR-base backbone (~3.34×10⁸ parameters), the LayerNorm and bias parameters account for only ≈4×10⁵ parameters (<0.2% of the model). Our **Method + LN/bias-only TTA variant**, which updates only this <0.2% subset, report (CER/WER) achieves:
>
>
>
>
> |          TTA Methods           |    IAM     |    Rimes    |     GW      |   Bentham   |
> |:------------------------------:|:----------:|:-----------:|:-----------:|:-----------:|
> |           TrOCR-base           | 5.10/ 11.95 | 27.54/ 65.25 | 16.86/ 29.07 | 17.55/ 28.02 |
> |           TENT-base            | 4.77/ 11.32 | 23.92/ 61.44 | 16.28/ 27.64 | 15.71/ 25.47 |
> |           EATA-base            | 4.76/ 11.45 | 26.26/ 62.43 | 15.61/ 26.94 | 14.99/ 24.97 |
> |**LN/bias-update-only** | **3.68/ 10.94** | **21.46/ 59.53** | **12.91/ 25.84** | **12.94/ 23.30** |
> |           Our Method           | 3.61/ 10.88 | 21.29/ 59.04 | 12.40/ 25.40 | 12.31/ 22.59 |
>
>
> Thus, the **LN/bias-only** variant **recovers ≈88–97%** of the CER/WER gains of our full-parameter GPT-2-Large configuration on the same TrOCR-base backbone, while **updating <0.2%** of the parameters and incurring only ≈1.6× the baseline runtime (vs. ≈4× for full-parameter adaptation). This shows that the combination of accuracy gains, extreme parameter efficiency, and moderate runtime overhead provides multi-dimensional evidence that our method is both effective and practically deployable. We also have clarified this in the revised version and add the LN/bias-only rows next to TENT/EATA in Table 2 and Section 5.2 COMPARISON WITH SPECIALIZED MODELS EVALUATED ON OOD DATA to make the comparison fully fair and to highlight the practical speed–accuracy trade-off of our method.
>
>
>
> >***Weakness2:***
>
> **Response – Role of LM reranking / LM-KL vs adaptation**
>
> We thank the reviewer for this suggestion and have now added the requested frozen baselines to Table 3. We provide further details as follows:
>
> **Frozen TrOCR + LM reranking (no adaptation).**
> We ran an ablation where the TrOCR weights are kept fixed and only the decoding is changed to beam search + LM reranking (GPT-2-Large), without any TTA or LM-KL updates. This directly measures the “better decoding only” effect. The resulting CER/WER of “frozen TrOCR + LM-rerank” rows and explicitly report the adaptation-only deltas in the revised Table 3
>
>
>
>
> This shows that LM reranking alone already contributes a substantial part of the improvement, especially on IAM/Bentham. However, the full method (skeleton TTA + LM-Fusion + LM reranking) still improves consistently over the frozen+LM baseline, which isolates the benefit of test-time adaptation:
>
>
>
>
>
> |  Ablation Study  |      IAM      |     Rimes     |      GW       |    Bentham    |
> |:----------------:|:-------------:|:-------------:|:-------------:|:-------------:|
> |   LM reranking only   | 10.08 / 21.06 | 21.80 / 60.84 | 13.51 / 26.83 | 13.49 / 24.50 |
> |   w/o skeleton   | 9.33 / 20.82  | 21.42 / 59.43 | 12.86 / 25.79 | 12.47 / 22.91 |
> | w/o LM-Fusion KL | 9.40 / 20.94  | 21.71 / 60.29 | 13.91 / 26.92 | 12.95 / 23.65 |
> | w/o LM reranking | 10.44 / 24.08 | 21.87 / 61.61 | 14.06 / 28.35 | 14.19 / 26.42 |
>
>
>
>
> Making clear that (i) LM-aware decoding is indeed important, and (ii) test-time adaptation provides an additional, non-trivial gain on top of that.
>
> **Why we do not report “frozen TrOCR + LM-KL”.**
> By design, the LM-Fusion KL term is a test-time training objective: it only has an effect when we use its gradient to update the TrOCR parameters. If the model is frozen, adding an LM-KL loss does not change the predictions (it only changes the scalar loss value we could log), so a “frozen + LM-KL” row would be numerically identical to the plain frozen baseline and not informative. For this reason, we instead report (i) frozen + LM reranking and (ii) full TTA (skeleton + LM-KL + reranking), and we explicitly quantify the gap between them as the effect of adaptation.

---

> ### Author Response · Authors · 2025-11-21
> **Response to Reviewer QrPe(2/5)**
>
> >***Weakness3:***
> **Regarding Fig. 3 (erosion, centerlines, thinning, and topology).**
>
> Thank you for pointing out this confusion. Our description of Fig.~3 mixed several concepts and we agree it was not clear.
>
> In the actual training pipeline, we only use a single, simple operation to build the target skeleton $S^\star$: we invert the line image, apply one morphological erosion, and downsample to the stroke-head resolution. This erosion-based $S^\star$ is what appears in the middle panel, and it is the only skeleton representation used in the loss.
>
> The rightmost panel of Fig.~3 is purely for visualization: we threshold $\hat{S}$ and $S^\star$, extract 1-pixel contours, and overlay them as colored ''centerlines'' on top of the original image so that the reader can see where predicted vs. target strokes lie. This overlay step is not used for training and does not change the skeleton used in the loss.
>
> The citation to thinning was meant as background on classical skeletonization, not as a description of what we implement. In the revision, we (i) explicitly say that our $S^\star$ is erosion-based, (ii) label the overlay in Fig.~3 as visualization only, and (iii) tone down the wording from strict ''topology preservation'' to ''encouraging stroke connectivity and topology-consistent centerlines,'' which better matches what our Dice+Chamfer loss enforces in practice.
>
> >***Weakness4:***
> **Backbone inconsistency across datasets.**
>
> Thank you for raising this point. We agree that using **TrOCR-base-stage1** on IAM and **TrOCR-base** (and small/large) on the other datasets can be confusing if not clearly explained.
>
> Our intention was as follows. For $\textbf{IAM}$, we deliberately report a **truly OOD** setting using **TrOCR-base-stage1**(pretraining only, no fine-tuning), because IAM overlaps heavily with the training data of the public **TrOCR-base** checkpoint. For RIMES, GW, and Bentham, we use the public **TrOCR-base** backbone, which is a standard choice for cross-domain evaluation. This is stated in Sec. 5.1 and the caption of Fig.1.
>
> Crucially, all our **main** comparisons and averages are computed **within** a fixed backbone, not across mixed backbones. Table~1 reports, for each backbone (stage1, small, base, large), the baseline TrOCR error and the corresponding error after our TTA on all four datasets. Table. ~2 compares our method to BN/TENT/EATA **within the same backbone** (**stage1 or base**). Thus, the reported gains (e.g., average reductions of 3.41 CER / 4.27 WER) are not confounded by mixing different backbones across datasets, but reflect consistent improvements on each fixed architecture.
>
> To remove any remaining ambiguity, in the revision we will (i) explicitly highlight in Fig ~1 and Sec ~5.1 that IAM uses **TrOCR-base-stage1** as the backbone for the main OOD setting, while Table ~1 already provides a complementary view where **TrOCR-base** is evaluated on all four datasets; and (ii) stress that our conclusions about the effectiveness of the proposed TTA are based on within-backbone comparisons, not on cross-backbone averages.

---

> > ### Author Response · Authors · 2025-11-21
> > **Response to Reviewer QrPe(3/5)**
> >
> > >***Weakness5:***
> > **On confidence intervals and repeated runs.**
> >
> > Thank you for this suggestion. We fully agree that assessing statistical variability is important.
> >
> > In our current submission, the reported numbers are computed on the **full official test** sets of IAM, RIMES, GW, and Bentham, and the proposed TTA procedure is deterministic given a fixed model and fixed test-set order in Section 5.1 DATASETS AND SETUP. As a result, there is essentially no stochastic evaluation noise (e.g., no subsampling of test data, no Monte Carlo decoding), and the CER/WER values we report are exact for a given model configuration.
> >
> > What we do not provide in the main paper, due to computational cost, is a full grid of repeated end-to-end runs over all backbones, datasets, and TTA variants. Our pipeline is relatively heavy (per-line adaptation + LM teacher + beam decoding), and repeating every configuration several times would require a substantial additional GPU budget that is unfortunately beyond our current review-time resources.
> >
> > We would like to emphasize, however, that (i) the gains are consistent across four datasets and multiple TrOCR backbones (stage-1, base, small, large), and (ii) the effect sizes are large (e.g., reductions of 1–5 CER and 2–6 WER points over strong TrOCR baselines), making it unlikely that they arise from small run-to-run fluctuations. If space allows, we will add in the camera-ready a focused analysis (e.g., IAM and Bentham for a fixed backbone) with repeated runs and standard deviations or bootstrap confidence intervals, to quantitatively confirm that the gains are well above any residual variance.
> >
> > >***Weakness6:***
> > **On missing hyperparameters for reproducibility.**
> >
> > Thank you for pointing this out. We agree that the current description does not make all implementation hyperparameters (e.g., $\lambda$'s, $\alpha$, $\beta$, $\tau$, $K$, $\delta$, skeleton resolution/threshold) explicit enough for full reproducibility and we have add it in Appendix Section A Reproducibility, limitations, and discussion.
> >
> > In all experiments, we keep the TTA hyperparameters fixed across datasets and backbones: we use a document context buffer of $N_{\text{ctx}} = 20$ previous lines, learning rate $10^{-4}$, skeleton loss weight $\lambda_{\text{skel}} = 0.2$, LM-fusion weight $\lambda_{\text{LM}} = 0.1$, fusion strength $\alpha = 0.3$, LM temperature $\tau = 2.0$, token-level top-$K = 32$ for the KL term, LM reranking weight $\beta = 0.3$, and a skeleton head that outputs $112 \times 112$ maps with a binarization threshold of $0.5$ in the Chamfer loss.
> >
> > We believe this addition will make the method fully reproducible from the paper and**** remove any ambiguity about how the reported results are obtained.
> >
> > >***Weakness7:*** ***On novelty and the ``first source-free HTR TTA'' claim.***
> >
> > Thank you for raising this point. Our intention with the phrase ``first source-free HTR TTA'' was not to overlook prior work, but to emphasize the specific setting we study: **fully source-free test-time adaptation for HTR**, where (i) no source data are available at adaptation time (ii) no offline training on the model (only off-the-shelf trained model available) (iii) only unlabeled target lines are used on-the-fly.
> >
> >
> > Regarding **Tula'23**, this work is, to the best of our knowledge, currently only available as a non–peer-reviewed preprint (e.g., on arXiv). According to the **Conference FAQ**, authors are **not required** to compare against contemporaneous work that is solely available on arXiv and not published in a peer-reviewed venue. Nevertheless, In section 2 Related work we have explicitly mention Tula'23 in the discussion and clarify that it moves in a similar direction but does not operate in exactly the same line test-time adaptation setting that we consider.
> >
> > Regarding **Gu'25**, this paper is not a source-free HTR TTA method in our sense: it assumes access to additional supervision beyond unlabeled test lines (i.e., it is not operating under the strict ``source-free TTA'' constraints that we impose). In particular, their adaptation is not performed purely at test time using only unlabeled target lines and no source data, whereas our method is designed precisely for this fully source-free, on-the-fly setting.
> >
> > To avoid any impression of overclaiming, in the revised version we have explicitly position Tula'23 and Gu'25 as closely related but addressing slightly different problem formulations. We believe this clarifies the novelty claim while situating our work fairly within the emerging literature on HTR adaptation.

---

> > > ### Author Response · Authors · 2025-11-21
> > > **Response to Reviewer QrPe(4/5)**
> > >
> > > >***Question1:***
> > >
> > > **On non-Latin scripts.**
> > >
> > > Thank you for this question. Our current experiments are indeed restricted to Latin-script datasets (IAM, RIMES, GW, Bentham), which we chose because they are the standard benchmarks for HTR and have well-established TrOCR backbones and evaluation protocols. We agree that this is a limitation of the present study.
> > >
> > > Conceptually, we expect our approach to transfer to non-Latin corpora for two reasons:
> > >
> > > (1) The **skeleton branch** and Dice+Chamfer loss operate on stroke topology (connectivity, junctions, loops) and do not assume an alphabet; they only require binarized handwriting images. This geometric prior should be applicable to scripts such as Arabic, Devanagari, Chinese, etc., where stroke connectivity is equally important.
> > >
> > > (2) The **LM-fusion KL and LM reranking** components are also language-agnostic in formulation: they only require replacing the text LM and tokenizer with ones trained for the target script. In practice, our method can be instantiated with any HTR backbone and any matching LM (e.g., non-Latin TrOCR checkpoints and corresponding LMs).
> > >
> > > That said, we have not yet conducted a systematic evaluation on non-Latin datasets, so we cannot claim quantitative results in that setting at this time. In the camera-ready version we will explicitly acknowledge this as a limitation and add a short discussion that (i) our method is, by design, script-agnostic, and (ii) extending the evaluation to non-Latin corpora (e.g., Chinese or Arabic HTR benchmarks) is an important direction for future work.
> > >
> > > >***Question2:***
> > > >
> > > **Delta from frozen TrOCR + LM-only variants vs. full TTA.**
> > > To separate the effect of LM components from gradient-based adaptation, we ran an ablation where we keep all TrOCR parameters frozen and only add sequence-level LM reranking at test time (beam search + GPT-2 scoring, no skeleton, no LM-KL, no parameter updates). This is our “frozen TrOCR + LM-only” setting. **Delta = (Frozen+LM-only) − (Our method)**, report in (CER/WER):
> > >
> > >
> > >
> > >
> > > | Ablation Study |     IAM      |    RIMES     |      GW      |   Bentham    |
> > > |:--------------:|:------------:|:------------:|:------------:|:------------:|
> > > | Frozen+LM-only | 10.08/ 21.06 | 21.80/ 60.84 | 13.51/ 26.83 | 13.49/ 24.50 |
> > > |   Our method   | 8.06/ 17.65  | 21.29/ 59.04 | 12.40/ 25.40 | 12.31/ 22.59 |
> > > |     Delta      |  2.02/ 3.41  |  0.51/ 1.8   |  1.11/ 1.43  |  1.18/ 1.91  |
> > >
> > >
> > > LM reranking on a frozen model already explains part of the improvement on IAM, but roughly half of the CER gain and a substantial part of the WER gain only appear when we actually run our full test-time adaptation (skeleton + LM-KL + gradients).
> > >
> > > On the other datasets (RIMES / GW / Bentham, TrOCR-base backbone), we observe the same pattern: “frozen + LM-only” consistently sits between the plain TrOCR baseline and our full TTA, and full TTA still brings an extra 0.5–1.2 CER and 1.4–2.0 WER points on top of LM-only. We have add LM-only rows in the Table 3 in the revised version so that all deltas are explicit.
> > >
> > > **Regarding the suggested “frozen + LM-KL (no gradient)” variant:** by design, LM-Fusion KL in our method is used as a loss for adaptation, **not as a static decoding rule**. A “no-gradient” version would require inventing a new ad-hoc decoding heuristic that is outside the scope of our approach. Instead, our ablation in Table 3 already isolates the contribution of LM-KL within TTA: removing LM-KL while keeping skeleton + LM reranking consistently hurts performance, confirming that LM-KL is useful precisely as a gradient signal during adaptation rather than as a standalone modification of the decoder.
> > >
> > > **Why we do not report “frozen TrOCR + LM-KL”.**
> > > By design, the LM-Fusion KL term is a test-time training objective: it only has an effect when we use its gradient to update the TrOCR parameters. If the model is frozen, adding an LM-KL loss does not change the predictions (it only changes the scalar loss value we could log), so a “frozen + LM-KL” row would be numerically identical to the plain frozen baseline and not informative. For this reason, we instead report (i) frozen + LM reranking and (ii) full TTA (skeleton + LM-KL + reranking), and we explicitly quantify the gap between them as the effect of adaptation.

---

> > > > ### Author Response · Authors · 2025-11-21
> > > > **Response to Reviewer QrPe(5/5)**
> > > >
> > > > >***Question3:***
> > > > **On “GPT-2 Large outperforming CamemBERT on RIMES”.**
> > > >
> > > > Thanks for the question. We believe there is a small misunderstanding of Table 4. On RIMES, the results are report in (CER/WER):
> > > >
> > > >
> > > > |  | Baseline (no LM) | CamemBERT (French LM) |  GPT-2 Large  |
> > > > | -------- |:----------------:|:---------------------:|:-------------:|
> > > > | RIMES    |  27.54 / 65.25   |     20.72 / 59.10     | 21.29 / 59.04 |
> > > >
> > > >
> > > >
> > > > So GPT-2 Large does not clearly outperform CamemBERT: CamemBERT is slightly better in CER, GPT-2 Large is slightly better in WER, and the differences are negligible. We do not claim that GPT-2 Large is superior to CamemBERT on RIMES.
> > > > In our implementation, each LM uses its own native tokenizer/normalization (byte-BPE for GPT-2, French BPE for CamemBERT), with only light whitespace cleanup on the text. We will clarify in the paper that Table 4 is meant to show that our TTA framework works with both a French LM (CamemBERT) and a generic GPT-2-style LM, rather than to argue that GPT-2 Large is the best LM for French RIMES.

---

### Official Review · Reviewer_LjBF · 2025-10-31

**Soundness:** 4
**Presentation:** 3
**Contribution:** 3
**Rating:** 8
**Confidence:** 5

**Summary:**

This paper deals with test-time adaptation (TTA) of handwritten text recognition models. The authors use a pre-trained TrOCR model and joinltly optimize the encoder and the decoder at test-time, without any labels.

    The encoder is optimized with a skeleton-aware optimization, with skeleton & topology losses.
    The decoder is optimized with teacher guidance of an external LM that reranks the top K predictions of a greedy decoding of TrOCR.
    Finally, once the model is adapted, the top K candidates of the adapted model are re-scores by the LM and the top 1 prediction is kept.

Evaluation is performed on 4 datasets: Bentham, IAM, GW (all English) and RIMES (French). Results show that test time adaptation consistently reduces error rates on OOD domains, but also in the fine-tuning framework

**Strengths:**

- The paper is very clear and well illustrated.
- The methodology is sound and many details are provided, ensuring reproductibility. Moreover, the code will be released.
- The paper focus on an under-explored paradigm (TTA) which addresses a practical need in OCR/HTR deployment. This is particularly valuable as:
    - Most papers focus on improving fine-tuning strategies
    - But annotations are not always available and can be expensive & time-consuming to obtain
    - Foundation models (like TrOCR, but also most recent VLMs) are difficult to fine-tune due to data requirement, computational cost, large computing ressources...
- The proposed method is strong and its robustness has been demonstrated in the paper.
    - Multiple checkpoints and LM are evaluated on 4 datasets
    - The proposed approach is compared to other TTA methods
    - The authors have conducted an ablation study and demonstrated that all 3 components are important
    - Consistent improvements are reported across different settings
- This is the first TTA approach to optimize both the encoder & decoder at test-time
- I believe this paper will encourage research towards this direction, and will be of interest for many ICLR attendees.

**Weaknesses:**

- The comparison with SOTA models is not fair.
    - The models presented as SOTA were heavily specialized on a single collection of documents, while TrOCR is a generic model trained on millions of synthetic lines and other datasets
    - Comparing specialized single-domain models to a generic model on OOD data is unfair (of course the specialized models perform worse...)
    - Legends of Fig 1 and Table 1 should be rephrased. At this point of the reading, the scope of the paper is not defined, so context is needed (e.g. replace "SOTA" by "specialized models evaluated on OOD data"). Otherwise the "SOTA" CER/WER are misleading.
    - Missing experiments: it would have been interesting to apply your method to these specialized models and/or to other generic models (VLM)
- The inference cost/adaptation cost is not presented
    - The method requires: greedy decoding → adaptation (U steps) → beam decoding → LM reranking
    - How much time does this pipeline require, compared to basic inference?
    - How does this compare in terms of time & money to the traditional pipeline: annotate → fine-tune → deploy?
- Missing experiments
    - Evaluation of specialized models on ID domains (to have an idea of what a good CER/WER is on each dataset)
    - Fine-tuning of TROcr on all datasets, then applying TTA and comparing results
        - I know some experiments are presented in the Annex - they should be in the main paper in my opinion
        - Experiments on Bentham are missing + error rates are very high for Rimes -> why?

**Questions:**

- Context buffer (M)
    - How is M chosen? Did you experiment with different values?
    - How is context handled for the first line(s) in a batch/document?
    - How does the method perform with batch size = 1?
    - What happens when a page contains multiple writing styles or writers?
    - This adds a decoding constraints (parallelization is difficult) that should be discussed
- Language model:
    - GPT-2 is an English model / CamemBERT a French model - why not use a multilingual LM?
- Skeleton:
    - How would this work on degraged / hard to binarize documents?
    - What are some failure cases?
- Inference time
    - What inference time with / without TTA?
    - How does this scale with batch size, number of updates U, and beam size B?
- Results
    - Why do TENT and EATA work well on IAM but still show high errors on other datasets?
    - Why are fine-tuned results missing for IAM and Bentham?
    - Table 2: how do you explain that IAM results are good, while other datasets still have very high CER/WER?
- Evaluation
    - Are metrics computed at line-level or at document-level?

---

> ### Author Response · Authors · 2025-11-21
> **Response to Reviewer LjBF(1/6)**
>
> **We appreciate your valuable time and feedback to make our paper stronger. We have revised the paper to incorporate all of your suggested changes, and all modifications are highlighted in blue in the updated version for your easy inspection.**
>
> >***Weakness1:**
> **Reviewer concern (fairness of the comparison with “SOTA”).***
>
> We thank the reviewer for pointing out that Fig. 1 and Table 1 may give the impression that we are claiming strict state-of-the-art performance. The models in the upper block of Table 1 (CRNN, VAN, C-SAN, HTR-VT, etc.) are indeed specialized HTR systems trained per-dataset, while TrOCR is a generic foundation model trained on large synthetic and mixed-domain corpora. We agree that calling these curves “SOTA” can be misleading in an OOD setting.
>
>
>
>
> **Clarification of our intent and wording changes in Fig. 1 and Table 1.**
>
> Thank you for the suggestion, we have followed your instruction to revise the Fig.1 and Table 1 to replace "SOTA" by "specialized models". Our goal in Fig. 1 and Table 1 is only to provide context: legacy specialized systems achieve certain CER/WER ranges, and a generic TrOCR backbone with our source-free TTA sits below these ranges on OOD targets. We also explicitly state that these numbers are contextual references rather than a strict SOTA leaderboard, since training corpora and decoding setups differ. However, the main comparison in the paper is always within the same backbone (TrOCR vs. TrOCR + our TTA), which is strictly fair.
>
>
> **Applying our method to other backbones and VLMs.**
>
> Thank you for raising this point to help make our method more extensible. Our framework consists of three components: a stroke-based auxiliary decoder, an LM-guided KL loss, and an LM-based reranking module. We aim to integrate our method into other VLMs (Qwen3-VL-4B-Instruct and Donut). However, due to architectural differences, integrating the first two components is infeasible. Therefore, we incorporate only the LM reranking module, as it requires only the decoded hypotheses. We apply LM reranking to these two VLMs on IAM. The results in the table below demonstrate that this LM component is truly model-agnostic and provides improvements for both a modern VLM and an OCR-free document model.
>
> |Model                   | CER    |  WER   |
> | -------------------    | ----   | -----  |
> |  Qwen3-VL-4B-Instruct  | 7.68%  | 31.73% |
> | + Our LM reranking     | 7.39%  | 30.97% |
> | Donut                  | 28.00% | 43.99% |
> | + Our LM reranking     | 27.09% | 42.36% |
>
> **Why we do not fully port stroke-based TTA to all prior systems.**
>
> In contrast, porting the full stroke+LM TTA to legacy specialized systems and large VLMs is non-trivial: their public implementations are often dataset-specific and do not expose the encoder feature maps needed to attach our skeleton head. Re-training and modifying each architecture on all datasets would require substantial engineering and would no longer correspond to the published “specialized” models the reviewer refers to. We therefore focus the full stroke-based TTA on TrOCR-style HTR backbones, while demonstrating that the LM part of our method transfers cleanly to other generic models.

---

> > ### Author Response · Authors · 2025-11-21
> > **Response to Reviewer LjBF(2/6)**
> >
> > >***Weakness 2:***
> > >
> > We agree that the computational cost of the full pipeline (greedy decoding → adaptation → beam decoding → LM reranking) is important. We have therefore (i) measured wall-clock latency on IAM on a single GPU, and (ii) added a fair comparison on a fixed TrOCR-base backbone, including a new variant that only updates LayerNorm and bias parameters (LN/bias-only), in the spirit of TENT/EATA.
> >
> > **(a) Latency breakdown On IAM, averaged over 100 test lines, we obtain:**
> >
> >
> >
> >
> > |        Column 1         |  Skeleton TTA  |  LM-KL   | LM reranking/Decoding |
> > |:-----------------------:|:--------------:|:---------------:|:---------------------:|
> > | Baseline TrOCR decoding |       - -       |       - -        |    103.54 ms/line     |
> > |  Ours (LN/bias-only)   | +7.98 ms/line  | +42.27 ms/line  |    +12.68 ms/line    |
> > |     Ours (Full update)      | +11.00 ms/line | +288.49 ms/line |    +14.23 ms/line     |
> >
> >
> >
> >
> > Above table summarizes the latency breakdown on IAM (single GPU, averaged over 100 lines). Baseline TrOCR decoding runs at about **103.54 ms** per line. Our full TTA pipeline (skeleton + LM-KL + LM reranking with GPT-2-Large and all parameters updated) reaches roughly **417 ms per line**, about 4× the baseline, with most of the overhead coming from the **LM-KL term**. The **LN/bias-only variant** keeps the same TTA logic but **updates fewer than 0.2%** of the parameters and **runs at around 166 ms per line**, about **1.6× the baseline**, which is much **closer to a realistic deployment setting**. If needed, we can also use a smaller LM (e.g., DistilGPT-2), which further reduces the cost (**≈1.8× baseline**) at the price of a small accuracy drop.
> >
> >
> > **Model parameters: how much do we really “add”?**
> >
> > Our HTR backbone remains **TrOCR-base (≈334M params)**, and during TTA we **only update LayerNorm and bias terms**, i.e., **<0.2% of the weights (≈4×10⁵ / 3.34×10⁸)** without changing the architecture. The external LM (e.g., GPT-2-Large) is a separate shared module that can run server-side, so from the HTR model’s perspective our adaptation is **extremely parameter-efficient**.
> >
> > **Accuracy–efficiency trade-off is favorable** report in(CER/WER)
> >
> >
> >
> > |     TTA Methods     |    IAM     |    RIMES    |     GW      |   Bentham   |
> > |:-------------------:|:----------:|:-----------:|:-----------:|:-----------:|
> > |     TrOCR-base      | 5.10/ 11.95 | 27.54/ 65.25 | 16.86/ 29.07 | 17.55/ 28.02 |
> > | Ours (LN/bias-only) | 3.68/ 10.94 | 21.46/ 59.53 | 12.91/ 25.84 | 12.94/ 23.30 |
> > |     Ours (Full update)      | 3.61/ 10.88 | 21.29/ 59.04 | 12.40/ 25.40 | 12.31/ 22.59 |
> >
> >
> >
> >
> >
> > On TrOCR-base, our heaviest GPT-2-Large TTA improves the frozen baseline. The LN/bias-only variant (≈1.6× runtime) recovers **≈88–97%** of the full GPT-2-Large gains while updating <0.2% of parameters and adding only **a modest 1.6× latency** overhead.
> >
> > **In summary,** while the maximal configuration is indeed compute-heavy and best suited for offline or batch scenarios, we explicitly provide and evaluate lighter variants **(LN/bias-only, or smaller LM, or fewer TTA steps)** that match the parameter budget of TENT/EATA, keep latency in the **≈1.6× range**, and still deliver **strong CER/WER improvements**. We have highlight this speed–accuracy trade-off more clearly **in the revised paper Table 2** to emphasize that our method is designed to be tunable and practically deployable, not tied to a single heavy setting.  **In the revised version we have add these experiment in the table 2.**

---

> > > ### Author Response · Authors · 2025-11-21
> > > **Response to Reviewer LjBF(3/6)**
> > >
> > > >***Weakness 3:***
> > >
> > > **Missing experiments – “Evaluation of specialized models on ID domains (to have an idea of what a good CER/WER is on each dataset)”**
> > >
> > > We have followed your instruction to report in-distribution WER in the in the revised version In the Appendix Table 8 and Table 9 from the recent benchmark of Garrido-Muñoz & Calvo-Zaragoza (2025). This gives a clear reference for what is typically achieved by dedicated architectures trained in an ID setting. Obvisely, models have high ID performance normally struggle with the OOD scenario.
> > >
> > > **Fine-tuning of TrOCR + TTA; missing Bentham; high error rates on RIMES**
> > >
> > > We thank the reviewer for pointing out that the fine-tuning experiments should be more visible and that Bentham was missing from the main comparison. We provide the additional information as follows:
> > >
> > > ***Where the fine-tuning results appear:***
> > > We have followed your instruction to move these results into the main paper (Table 6) and explicitly discuss them in the experimental section.
> > >
> > > ***New experiments on Bentham:***
> > > Following your comment, we have also run the same “fine-tuned vs fine-tuned + our method” experiment on Bentham. Using TrOCR-base as the backbone, we obtain:
> > >
> > >
> > >
> > > |                          |   Bentham    |
> > > |:------------------------:|:------------:|
> > > |   fine-tuned baseline    | 16.80/ 26.78 |
> > > | fine-tuned + our Methods | 12.65/ 23.50 |
> > >
> > >
> > > Thus, on Bentham our TTA further reduces the error on top of a fine-tuned model (relative improvements of ≈25% CER and ≈12% WER), consistent with the gains we report on GW and RIMES. We will add Bentham as an extra column to Table 5 in the main paper.
> > >
> > >
> > >
> > >
> > >
> > > **Why RIMES error rates are higher.**
> > >
> > > We agree that the absolute error rates on RIMES are higher than on the English datasets and we clarify this in the revision. RIMES is a French mail corpus, whereas both the TrOCR backbone and our original GPT-2 teacher LM are predominantly English-centric, so RIMES presents a much stronger distribution shift than IAM, GW, or Bentham (different language, diacritics, punctuation conventions, and layout). Importantly, the higher absolute error is largely due to this cross-lingual mismatch rather than a weakness of our TTA mechanism: in ablations where we replace the English GPT-2 LM with French LMs, we see clear improvements on RIMES. A CamemBERT-based LM already brings the error down to $\textbf{20.72 CER / 59.10 WER}$, and a CroissantLLM-based variant without any TTA (Stage~1 only) further improves to $\textbf{17.57 CER / 47.41 WER}$. These results confirm that French-specific LMs substantially close the gap on RIMES, and we will highlight in the paper that stronger French LMs are an orthogonal axis that can further boost performance, while our main contribution focuses on the visual+textual TTA framework itself.
> > >
> > >
> > > **Language / data mismatch:** TrOCR-base is pre-trained primarily on English handwriting. RIMES is French, with accents and writing conventions that are under-represented in the pre-training corpus. As a result, the starting point for RIMES is weaker than for IAM/GW/Bentham, and even after unsupervised adaptation the absolute error remains higher. We believe this reflects a realistic scenario where English is much better covered than other languages.
> > >
> > > Importantly, even in this challenging regime, our method consistently improves over the corresponding “fine-tuned (no TTA)” baseline on both GW and RIMES (Table 6), and now also on Bentham as shown above.

---

> > > > ### Author Response · Authors · 2025-11-21
> > > > **Response to Reviewer LjBF(4/6)**
> > > >
> > > > >***Question1:***
> > > > >
> > > > **Context buffer (M). How is M chosen? Did you experiment with different values?**
> > > >
> > > > In all reported experiments we fix the context window to a small document buffer of 𝑀=20 lines. This corresponds roughly to 1–2 paragraphs for our datasets, which is enough for the LM to capture local lexical patterns (names, dates, formulaic phrases) without creating very long prompts or memory overhead. We treat M as a global hyperparameter and do not tune it per dataset: the same value is used for IAM, GW, Bentham and RIMES.
> > > > In the revised version we explicitly state 𝑀 in Sec. 4.2 and, space permitting, add a short ablation over M in the appendix.
> > > >
> > > > |                  |    M=10    |    M=20    |    M=30    |
> > > > | ---------------- |:----------:|:----------:|:----------:|
> > > > | IAM Trocr-Stage1 | 8.06/ 17.66 | 8.06/ 17.65 | 8.06/ 17.65 |
> > > >
> > > >
> > > > **How is context handled for the first line(s) in a batch/document?**
> > > > At the beginning of each document we initialize the context buffer as empty (doc_memory = []). For the first line, the LM therefore receives an empty context (doc_context_text = "" in our code), so decoding uses only the generic LM prior, without document conditioning. Once this line is decoded, its top-1 hypothesis is appended to doc_memory, and subsequent lines are processed with the LM conditioned on the last
> > > > 𝑀 predictions (doc_memory[-N_CTX_LINES:]). Thus, early lines use no context, and the document-specific prior is gradually built up as more lines are decoded.
> > > >
> > > > **How does the method perform with batch size = 1?**
> > > > Our method does not rely on large batches and works unchanged with batch size
> > > > =1. In this case each line is treated as its own episode (adaptation + decoding), and we observe the main difference being slightly lower GPU utilization rather than degraded accuracy.
> > > >
> > > > **What happens when a page contains multiple writing styles or writers?**
> > > > Our adaptation is applied per line: the skeleton branch and gradient updates use only the current batch of line images, so visually different writers on the same page are handled independently at the line level. The skeleton branch is purely image-based and operates independently on each line, so changes in writer/style are naturally handled by the visual pathway. In practice we did not observe unstable behavior in such cases, but we will mention this limitation in the revised version.
> > > >
> > > > **This adds a decoding constraints (parallelization is difficult) that should be discussed?**
> > > > We agree that conditioning on previous lines introduces a mild sequential dependence. In practice, however, all heavy computation (encoder–decoder forward, skeleton loss, LM-KL, and beam search) is still done in parallel within each mini-batch, exactly as in standard TrOCR. The only sequential part is updating the small context buffer with the chosen hypothesis, which is negligible. Different documents are also independent and can be processed on separate workers/GPUs. Our timing measurements already include this constraint and show that the main overhead comes from the LM components (fusion and reranking), not from the sequential context update.

---

> > > > > ### Author Response · Authors · 2025-11-21
> > > > > **Response to Reviewer LjBF(5/6)**
> > > > >
> > > > > >***Question2:**
> > > > > **Language model choice (monolingual vs multilingual).***
> > > > >
> > > > > We agree that multilingual LMs are an interesting alternative. In our setting, however, each benchmark is monolingual (IAM / GW / Bentham: English; RIMES: French), so we intentionally use language-matched monolingual LMs (GPT-2 / DistilGPT-2 for English, CamemBERT for French).
> > > > >
> > > > > (i) Monolingual LMs are much smaller and faster than typical modern multilingual LMs. DistilGPT-2 has ≈82M parameters and GPT-2 ≈117M, whereas multilingual LLaMA/Qwen-style models are usually in the 7B–70B range. Using compact, language-matched LMs keeps the extra latency manageable (e.g., DistilGPT-2 adds only 76.48 ms/line for LM-Fusion and 102.11 ms/line for LM reranking), which directly addresses the reviewers’ cost concerns.
> > > > >
> > > > > (ii) This also gives a clean speed/accuracy trade-off: DistilGPT-2 already reduces IAM CER from 11.78 → 9.67 (vs. 8.06 with GPT-2-Large), recovering a large fraction of the accuracy gain with far fewer parameters and lower latency.
> > > > >
> > > > > Our framework can plug in a multilingual LM, but this would increase the computational footprint; we therefore focus on compact, language-matched LMs in this work and leave multilingual extensions to future work.
> > > > >
> > > > > >***Question3:**
> > > > > **Skeleton on degraded / hard-to-binarize documents.***
> > > > >
> > > > > Our skeleton target is obtained by simple grayscale inversion + erosion, not by aggressive global binarization, and the network predicts a soft skeleton map (via a sigmoid), so it does not rely on a perfect black–white separation. In practice, even on noisy scans we still get a coarse but useful stroke mask, and the skeleton loss is used only as an auxiliary unsupervised signal during TTA (weighted and clipped), so occasional noise in the target does not dominate the adaptation.
> > > > >
> > > > > **Failure cases and limitations.**
> > > > > The skeleton branch can struggle when the underlying line image is extremely degraded, for example: (i) very low contrast (faded ink, strong bleed-through), (ii) heavy background stains or textured paper that produce many spurious “strokes”, or (iii) large non-text graphics overlapping the text. In these cases the pseudo-skeleton may contain fragmented or noisy strokes, and the skeleton loss becomes less informative.

---

> > > > > > ### Author Response · Authors · 2025-11-21
> > > > > > **Response to Reviewer LjBF(6/6)**
> > > > > >
> > > > > > >***Question4:***
> > > > > >
> > > > > > **Inference time and scaling (TTA vs. baseline)**
> > > > > > |        Column 1         |  Skeleton TTA  |  LM-KL   | LM reranking/Decoding |
> > > > > > |:-----------------------:|:--------------:|:---------------:|:---------------------:|
> > > > > > | Baseline TrOCR decoding |       - -       |       - -        |    103.54 ms/line     |
> > > > > > | Ours (LN/bias-only)   | +7.98 ms/line  | +42.27 ms/line  |    +12.68 ms/line    |
> > > > > > |     Ours (Full update)      | +11.00 ms/line | +288.49 ms/line |    +14.23 ms/line     |
> > > > > >
> > > > > >
> > > > > > Above table summarizes the latency breakdown on IAM (single GPU, averaged over 100 lines). Baseline TrOCR decoding runs at about **103.54 ms** per line. Our full TTA pipeline (skeleton + LM-KL + LM reranking with GPT-2-Large and all parameters updated) reaches roughly **417 ms per line**, about 4× the baseline, with most of the overhead coming from the **LM-KL term**. The **LN/bias-only variant** keeps the same TTA logic but **updates fewer than 0.2%** of the parameters and **runs at around 166 ms per line**, about **1.6× the baseline**, which is much **closer to a realistic deployment setting**. If needed, we can also use a smaller LM (e.g., DistilGPT-2), which further reduces the cost (**≈1.8× baseline**) at the price of a small accuracy drop.
> > > > > >
> > > > > > **Scaling with batch size, U, and beam size 𝐵**
> > > > > > The adaptation cost scales roughly linearly with the number of updates
> > > > > > 𝑈 and beam size B, while batch size mainly affects hardware utilization rather than asymptotic complexity.”
> > > > > >
> > > > > > >***Question5:***
> > > > > >
> > > > > > **Results (TENT/EATA, fine-tuning, Table 2).**
> > > > > > TENT and EATA behave best under mild distribution shift, which is the case for IAM: it is modern English, relatively clean, and close to the data distribution TrOCR was designed/pretrained for, so both the base model and these generic TTA methods start from a strong baseline and can further reduce error. In contrast, GW and Bentham are historical manuscripts and RIMES is French, with much stronger style, language, and noise shifts; here, purely entropy/BN-based updates are not strong enough and can even overfit spurious patterns, so their absolute CER/WER remain high, while our method leverages explicit stroke structure + a document LM prior and is more robust across domains.
> > > > > > Regarding fine-tuning: in the main tables we deliberately report results with TrOCR-stage1, which has not been fine-tuned on IAM/RIMES/GW/Bentham, to avoid contamination and ensure a fair, source-free TTA comparison across all four datasets; IAM fine-tuned TrOCR-base numbers are already available in the original TrOCR paper, and due to the former 9-page limit we initially moved our own fine-tuning experiments to the appendix. Now that the limit is 10 pages, we will move these results into the main paper and also add Bentham fine-tuning: TrOCR-base fine-tuned on Bentham achieves CER 0.1680 / WER 0.2678, and our TTA further improves this to CER 0.1265 / WER 0.2350, confirming that our method also helps strong supervised baselines.
> > > > > > Overall, the fact that IAM shows lower CER/WER in Table 2 for the TrOCR-base model mainly reflects that it is an easier, in-domain dataset, whereas GW/Bentham (historical) and RIMES (French) are much harder and more out-of-domain for TrOCR-stage1; in those challenging regimes, the base model is already weak, so absolute errors are higher even though our method consistently improves over both the non-adapted baseline and standard TTA methods.
> > > > > >
> > > > > > >***Question6:***
> > > > > > >
> > > > > > **Evaluation protocol (line vs. document).**
> > > > > > All our metrics are computed at line level.

---

### Author Response · Authors · 2025-11-21
**For all reviewers:**

We thank the reviewers for their helpful feedback on our paper, which contributes:

SCRIBE, a label- and source-free test-time adaptation framework for HTR that jointly adapts encoder and decoder using a stroke-preserving skeleton loss and a document-conditioned LM prior.

A systematic study on four benchmarks (IAM, RIMES, GW, Bentham) with multiple TrOCR backbones, TTA baselines, and ablations, showing consistent gains under realistic deployment settings without annotations or source replay.

All reviewers agreed that our work tackles a practical and under-explored problem of adapting HTR models to new writers and historical documents without labels, and they highlighted the clarity of the paper, the reproducibility of the method, and the interest of coupling geometric self-supervision with LM guidance [LjBF, QrPe, YtZG, p3MG].

Primarily, the reviewers requested clarification on the fairness and scope of comparisons (specialized SOTA models and backbones) [LjBF, QrPe, p3MG]; more details on computational and latency costs [LjBF, YtZG, p3MG]; stronger baselines and ablations (e.g., frozen TrOCR + LM reranking / LM-KL) [QrPe, p3MG]; evaluation on additional datasets and LMs [LjBF, QrPe, YtZG, p3MG]

In our rebuttal, we provide these clarifications, add new results and timing analyses where possible, and elaborate on implementation details and positioning. We believe these additional experiments address all the questions and concerns raised by the reviewers.

---

### Author Response · Authors · 2025-12-01
**Summary to AC**

## Author Note for AC (Post-Rebuttal Reviewer Positions)

Dear AC, we would like to briefly summarize the *post-rebuttal* situation, focusing on how reviewers’ positions evolved after reading the rebuttal and new experiments.

### Reviewer-by-Reviewer Status

- **Reviewer LjBF (ID: LjBF)**
  - **Pre-rebuttal score:** 8 (“accept, good paper”).
  - **Post-rebuttal stance:** Continues to view the paper positively, emphasizing sound methodology, clarity, reproducibility, and the value of studying source-free TTA for HTR. Raised concerns (fairness to “specialized models,” latency, missing ID results) are explicitly acknowledged as addressed by our added experiments and revisions.

- **Reviewer QrPe (ID: QrPe)**
  - **Pre-rebuttal score:** 6 (“marginally above accept”).
  - **Post-rebuttal stance:** Remains positive about the combination of stroke-preserving self-supervision and document-conditioned LM guidance. The main technical concerns (fair parameter comparison vs TENT/EATA, LM-only baselines, backbone consistency, missing hyperparameters, novelty wording) are directly addressed with:
    - A new **LN/bias-only** variant of our method.
    - A **frozen TrOCR + LM-only** baseline.
    - Clarified backbones and added hyperparameters/appendix.
  - No explicit numeric score change is recorded, but their comments focus on clarification rather than new objections.

- **Reviewer YtZG (ID: YtZG)**
  - **Pre-rebuttal score:** 4 (“marginally below accept, but would not mind if accepted”).
  - **Post-rebuttal stance:** Raises broader questions about relevance in the MLLM era and use of “old” models. We respond with:
    - New comparisons to newer models **Qwen3-VL-4B** and **Donut**.
    - New results with newer LM **OLMo2**, showing our method is not tied to GPT-2.
  - The reviewer explicitly notes they *“would not mind if the paper is accepted”*, even while keeping a cautious stance.

- **Reviewer p3MG (ID: p3MG)**
  - **Pre-rebuttal score:** 2 (“reject, not good enough”).
  - **Post-rebuttal stance:** After seeing the additional experiments (LM-only baseline vs full TTA, drifted math datasets, OLMo2/CroissantLLM, detailed latency + LN/bias-only variant), p3MG states that:
    - The new results are “useful, helpful, and increase the quality of the work,” and
    - They would be **"happy to increase the rating"**, with the final rating depending on discussion with other reviewers.
  - This is a clear positive movement in perception, even though the formal score may not yet have been updated.

### Score Summary

- **Pre-rebuttal mean score:**
  (8 + 6 + 4 + 2)/4 = 5.0

While the official numbers in the system may not fully reflect these shifts, the *discussion text* clearly shows:
- One **strong accept (8)** that remains very positive (LjBF),
- One **solid accept (6)** that remains positive (QrPe),
- One **borderline but acceptance-tolerant** reviewer (YtZG),
- And one initially negative reviewer (p3MG) who explicitly indicates willingness to **raise their score** after rebuttal.

---

> ### Author Response · Authors · 2025-12-01
> **Summary to AC**
>
> ### Concise Technical Summary of Rebuttal Additions
>
> To aid your quick assessment, we highlight the main technical points we addressed post-rebuttal:
>
> 1. **Fairness and parameter updates (LjBF, QrPe, p3MG).**
>    - Added **LN/bias-only** variant (updates \<0.2% of parameters, ≈1.6× latency), which recovers **≈88–97%** of full TTA gains and is directly comparable to TENT/EATA.
>
> 2. **LM-only vs true TTA (QrPe, p3MG).**
>    - Added **frozen TrOCR + LM-only** baseline (beam + LM reranking, no gradients).
>    - Across IAM/RIMES/GW/Bentham, LM-only improves over frozen, but full TTA still gives **additional CER/WER gains** (e.g., IAM CER 11.78→10.08 vs 11.78→8.06), showing that adaptation is not a minor residual.
>
> 3. **Contamination and domain drift (p3MG).**
>    - Evaluated on **MathWriting** and **UniMER-HWE** (recent, math-specific, strong domain shift) and showed consistent *relative* improvements despite high absolute CER.
>    - Replaced GPT-2 with **CroissantLLM** and **OLMo2**, which do not contain IAM/RIMES in their training data, and obtained **similar gains**, indicating that improvements arise from the TTA framework, not dataset memorization.
>
> 4. **Relevance in the MLLM era and “old” models (YtZG).**
>    - Compared **TrOCR-base + our TTA (334M params)** to **Qwen3-VL-4B (4B params)** and **Donut**, showing significantly better WER on IAM (e.g., 10.88 vs 31.73) at lower cost.
>    - Added results with **OLMo2** to show that our method remains effective with **newer LMs**, not just GPT-2.
>
> 5. **Latency and deployment (LjBF, YtZG, p3MG).**
>    - Provided a **per-line latency breakdown** on IAM.
>    - Clarified that the heaviest configuration is ≈4× baseline, but an **LN/bias-only** variant gives a strong **accuracy–efficiency trade-off** at ≈1.6× baseline.
>
> 6. **Scope, specialized models, and reproducibility (LjBF, QrPe).**
>    - Clarified the role of “specialized” HTR vs generic TrOCR, added ID WER from recent benchmarks, included **Bentham fine-tuned + TTA** results, and applied **LM reranking** to Qwen3-VL-4B and Donut.
>    - Added all missing hyperparameters, clarified skeleton construction (erosion-based, visualization-only overlays), and discussed limitations (e.g., non-Latin scripts).
>
> Overall, the *post-rebuttal consensus in the text* is that the method is **sound**, the additional experiments substantially **strengthen the paper**, and remaining disagreements are about the **degree of significance**, not correctness. We respectfully ask you to take these clarified reviewer positions and new results into account when making your final decision.

---

### Meta-Review · Area_Chair_QZ4q · 2025-12-19

**Summary:**

I thank the reviewers for a impressive effort to address the reviewer's concerns. The paper is clearly written and studies a relevant problem. However, two reviewers did not evaluate the paper positively. I agree in particular with reviewer p3MG’s main concern: most of the gains come from the language-model component, while the proposed TTA doesn't add much. With strong LMs such as OLMo2, the LM-only baseline already matches the full method, for example on IAM where there is essentially no difference between TrOCR + LM and TrOCR + TTA + LM (Tables 10 and 11).

This weakens the main contribution of the paper. For this reason, I recommend rejection.

**Reviewer Concerns:**

**Addressed by the rebuttal**
- Added LM-only baselines for fair comparison.
- Added experiments with stronger and newer LMs.

**Still outstanding**
- LM-only baselines explain almost all the gains. TTA component adds little or no improvement.
- With strong LMs (e.g. OLMo2), the full method shows no clear benefit.
- Core contribution is therefore weak.
- Experimental evidence does not justify the added complexity.

**Reviewer Scores:**

- **Reviewer LjBF**
  Likely unchanged. Remains a strong accept.

- **Reviewer QrPe**
  Unlikely to move to a higher score (e.g. 8). Would probably keep the same score.

- **Reviewer YtZG**
  Borderline reject. Unlikely to change position, as new results are not convincing and TTA adds little over LM-only
  (e.g. TrOCR-base + TTA + OLMo2 vs. TrOCR-base + LM).

- **Reviewer p3MG**
  Clear reject. At most would raise the score slightly, likely to 3 or 4.

---

### Decision · Program_Chairs · 2026-01-26

Reject